# SOFTS: Efficient Multivariate Time Series Forecasting with Series-Core Fusion

**Lu Han,**[*] **Xu-Yang Chen,**[*] **Han-Jia Ye,**[†] **De-Chuan Zhan**
School of Artificial Intelligence, Nanjing University, China
National Key Laboratory for Novel Software Technology, Nanjing University, China
{hanlu, chenxy, yehj, zhandc}@lamda.nju.edu.cn

## Abstract

Multivariate time series forecasting plays a crucial role in various fields such as finance, traffic management, energy, and healthcare. Recent studies have highlighted the advantages of channel independence to resist distribution drift but neglect channel correlations, limiting further enhancements. Several methods utilize mechanisms like attention or mixer to address this by capturing channel correlations, but they either introduce excessive complexity or rely too heavily on the correlation to achieve satisfactory results under distribution drifts, particularly with a large number of channels. Addressing this gap, this paper presents an efficient MLP-based model, the Series-cOre Fused Time Series forecaster (SOFTS), which incorporates a novel STar Aggregate-Redistribute (STAR) module. Unlike traditional approaches that manage channel interactions through distributed structures, *e.g.*, attention, STAR employs a centralized strategy to improve efficiency and reduce reliance on the quality of each channel. It aggregates all series to form a global core representation, which is then dispatched and fused with individual series representations to facilitate channel interactions effectively. SOFTS achieves superior performance over existing state-of-the-art methods with only linear complexity. The broad applicability of the STAR module across different forecasting models is also demonstrated empirically. We have made our code publicly available at `https://github.com/Secilia-Cxy/SOFTS`.

## 1 Introduction

Time series forecasting plays a critical role in numerous applications across various fields, including environment [9], traffic management [16], energy [17], communication [45], and healthcare [28]. The ability to accurately predict future values based on previously observed data is fundamental for decision-making, policy development, and strategic planning in these areas. Historically, models such as ARIMA and Exponential Smoothing were standard in forecasting, noted for their simplicity and effectiveness in certain contexts [2]. However, the emergence of deep learning models, particularly those exploiting structures like Recurrent Neural Networks (RNNs) [14, 3, 30] and Convolutional Neural Networks (CNN) [1, 8], has shifted the paradigm towards more complex models capable of understanding intricate patterns in time series data. To overcome the inability to capture long-term dependencies, Transformer-based models have been a popular direction and achieved remarkable performance, especially on long-term multivariate time series forecasting [51, 29, 27].

Earlier on, Transformer-based methods perform embedding techniques like linear or convolution layers to aggregate information from different channels, then extract information along the temporal dimension via attention mechanisms [51, 37, 52]. However, such channel mixing structures were

---

[*]Equal Contribution
[†]Corresponding Author

38th Conference on Neural Information Processing Systems (NeurIPS 2024).

found vulnerable to the distribution drift, to the extent that they were often less effective than simpler methods like linear models [48, 11]. Consequently, some studies adopted a channel-independence strategy and achieved favorable results [29, 24, 36]. Yet, these methods overlooked the correlation between channels, thereby hindering further improvements in model performance. Subsequent studies captured this correlation information through mechanisms such as attention, achieving better outcomes, and demonstrating the necessity of information transfer between channels [50, 35, 27]. However, these approaches either employed attention mechanisms with high complexity [27] or struggled to achieve state-of-the-art (SOTA) performance [7]. Therefore, effectively integrating the robustness of channel independence and utilizing the correlation between channels in a simpler and more efficient manner is crucial for building better time series forecasting models.

In response to these challenges, this study introduces an efficient MLP-based model, the Series-cOre Fused Time Series forecaster (SOFTS), designed to streamline the forecasting process while also enhancing prediction accuracy. SOFTS first embeds the series on multiple channels and then extracts the mutual interaction by the novel STar Aggregate-Redistribute (STAR) module. The STAR at the heart of SOFTS ensures scalability and reduces computational demands from the common quadratic complexity to only linear. To achieve that, instead of employing a distributed interaction structure, STAR employs a centralized structure that first gets the global core representation by aggregating the information from different channels. Then the local series representation is fused with the core representation to realize the indirect interaction between channels. This centralized interaction not only reduces the comparison complexity but also takes advantage of both channel independence and aggregated information from all the channels that can help improve the local ones [42]. Our empirical results show that our SOFTS method achieves better results against current state-of-the-art methods with lower computation resources. Besides, SOFTS can scale to time series with a large number of channels or time steps, which is difficult for many methods based on Transformer without specific modification. Last, the newly proposed STAR is a universal module that can replace the attention in many models. Its efficiency and effectiveness are validated on various current transformer-based time series forecasters. Our contributions are as follows:

1. We present Series-cOre Fused Time Series (SOFTS) forecaster, a simple MLP-based model that demonstrates state-of-the-art performance with lower complexity.

2. We introduce the STar Aggregate-Redistribute (STAR) module, which serves as the foundation of SOFTS. STAR is designed as a centralized structure that uses a core to aggregate and exchange information from the channels. Compared to distributed structures like attention, the STAR not only reduces the complexity but also improves robustness against anomalies in channels.

3. Lastly, through extensive experiments, the effectiveness and scalability of SOFTS are validated. The universality of STAR is also validated on various attention-based time series forecasters.

## 2    Related Work

**Time series forecasting.**    Time series forecasting is a critical area of research that finds applications in both industry and academia. With the powerful representation capability of neural networks, deep forecasting models have undergone a rapid development [23, 40, 39, 4, 5, 15, 32]. Two widely used methods for time series forecasting are recurrent neural networks (RNNs) and convolutional neural networks (CNNs). RNNs model successive time points based on the Markov assumption [14, 3, 30], while CNNs extract variation information along the temporal dimension using techniques such as temporal convolutional networks (TCNs) [1, 8]. However, due to the Markov assumption in RNN and the local reception property in TCN, both of the two models are unable to capture the long-term dependencies in sequential data. Recently, the potential of Transformer models for long-term time series forecasting tasks has garnered attention due to their ability to extract long-term dependencies via the attention mechanism [51, 37, 52].

**Efficient long-term multivariate forecasting and channel independence.**    Long-term multivariate time series forecasting is increasingly significant in decision-making processes [9]. While Transformers have shown remarkable efficacy in various domains [34], their complexity poses challenges in long-term forecasting scenarios. Efforts to adapt Transformer-based models for time series with reduced complexity include the Informer, which utilizes a probabilistic subsampling strategy for more efficient attention mechanisms [51], and the Autoformer, which employs autocorrelation and fast

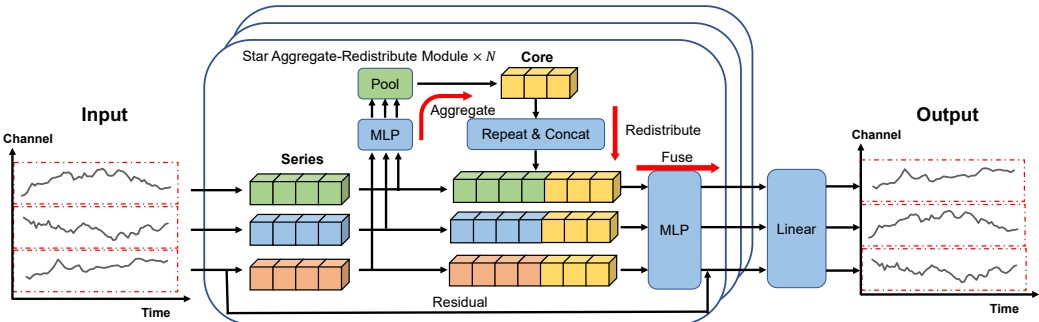

Figure 1: Overview of our SOFTS method. The multivariate time series is first embedded along the temporal dimension to get the series representation for each channel. Then the channel correlation is captured by multiple layers of STAR modules. The STAR module utilizes a centralized structure that first aggregates the series representation to obtain a global core representation, and then dispatches and fuses the core with each series, which encodes the local information.

Fourier transforms to expedite computations [37]. Similarly, FEDformer applies attention within the frequency domain using selected components to enhance performance [52]. Despite these innovations, models mixing channels in multivariate series often exhibit reduced robustness to adapt to distribution drifts and achieve subpar performance [48, 11]. Consequently, some researchers have adopted a channel-independent approach, simplifying the model architecture and delivering robust results as well [29, 24]. However, ignoring the interactions among variates can limit further advancements. Recent trends have therefore shifted towards leveraging attention mechanisms to capture channel correlations [50, 35, 27]. Even though the performance is promising, their scalability is limited on large datasets. Another stream of research focuses on modeling time and channel dependencies through simpler structures like MLP [49, 7, 44]. Yet, they usually achieve sub-optimal performance compared to SOTA transformer-based methods, especially when the number of channels is large.

In this paper, we propose a new MLP-based method that breaks the dilemma of performance and efficiency, achieving state-of-the-art performance with merely linear complexity to both the number of channels and the length of the lookback window.

## 3 SOFTS

Multivariate time series forecasting (MTSF) deals with time series data that contain multiple variables, or channels, at each time step. Given historical values $X \in \mathbb{R}^{C \times L}$ where $L$ represents the length of the lookback window, and $C$ is the number of channels. The goal of MTSF is to predict the future values $Y \in \mathbb{R}^{C \times H}$, where $H > 0$ is the forecast horizon.

### 3.1 Overview

Our Series-cOre Fused Time Series forecaster (SOFTS) comprises the following components and its structure is illustrated in Figure 1.

**Reversible instance normalization.** Normalization is a common technique to calibrate the distribution of input data. In time series forecasting, the local statistics of the history are usually removed to stabilize the prediction of the base forecaster and restore these statistics to the model prediction [18]. Following the common practice in many state-of-the-art models [29, 27], we apply reversible instance normalization which centers the series to zero means, scales them to unit variance, and reverses the normalization on the forecasted series. For PEMS dataset, we follow Liu et al. [27] to selectively perform normalization according to the performance.

**Series embedding.** Series embedding is an extreme case of the prevailing patch embedding in time series [29], which is equivalent to setting the patch length to the length of the whole series [27]. Unlike patch embedding, series embedding does not produce extra dimension and is thus less complex than patch embedding. Therefore, in this work, we perform series embedding on the lookback window.

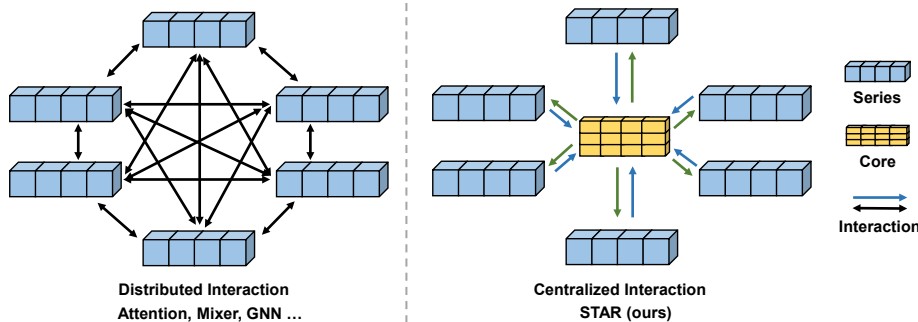

Figure 2: The comparison of the STAR module and several common modules, like attention, GNN and mixer. These modules employ a distributed structure to perform the interaction, which relies on the quality of each channel. On the contrary, our STAR module utilizes a centralized structure that first aggregates the information from all the series to obtain a comprehensive core representation. Then the core information is dispatched to each channel. This kind of interaction pattern reduces not only the complexity of interaction but also the reliance on the channel quality.

Concretely, we use a linear projection to embed the series of each channel to $\boldsymbol{S}_0 = \mathbb{R}^{C \times d}$, where $d$ is the hidden dimension:

$$\boldsymbol{S}_0 = \text{Embedding}(\boldsymbol{X}). \tag{1}$$

**Channel interaction.** The series embedding is refined by multiple layers of STAR modules:

$$\boldsymbol{S}_i = \text{STAR}(\boldsymbol{S}_{i-1}), \quad i = 1, 2, \ldots, N. \tag{2}$$

The STAR module utilizes a star-shaped structure that exchanges information between different channels, as will be fully described in the next section.

**Linear predictor.** After $N$ layers of STAR, we use a linear predictor ($\mathbb{R}^d \mapsto \mathbb{R}^H$) to produce the forecasting results. Assume the output series representation of layer $N$ to be $\boldsymbol{S}_N$, the prediction $\hat{\boldsymbol{Y}} \in \mathbb{R}^{C \times H}$ is computed as:

$$\hat{\boldsymbol{Y}} = \text{Linear}(\boldsymbol{S}_N).$$

## 3.2 STar Aggregate-Redistribute Module

Our main contribution is a simple but efficient STar Aggregate-Redistribute (STAR) module to capture the dependencies between channels. Existing methods employ modules like attention to extract such interaction. Although these modules directly compare the characteristics of each pair, they are faced with the quadratic complexity related to the number of channels. Besides, such a distributed structure may lack robustness when there are abnormal channels for the reason that they rely on the extract correlation between channels. Existing research on channel independence has already proved the untrustworthy correlations on non-stationary time series [48, 11]. To this end, we propose the STAR module to solve the inefficiency of the distributed interaction modules. This module is inspired by the star-shaped centralized system in software engineering, where instead of letting the clients communicate with each other, there is a server center to aggregate and exchange the information [31, 10], whose advantage is efficient and reliable. Following this idea, the STAR replaces the mutual series interaction with the indirect interaction through a core, which represents the global representation across all the channels. Compared to the distributed structure, STAR takes advantage of the robustness brought by aggregation of channel statistics [11], and thus achieves even better performance. Figure 2 illustrates the main idea of STAR and its difference between existing models like attention [34], GNN [20] and Mixer [33].

Given the series representation of each channel as input, STAR first gets the core representation of the multivariate series, at the heart of our SOFTS method. We define the core representation as follows:

**Definition 3.1** (Core Representation). Given a multivariate series with $C$ channels $\{\boldsymbol{s}_1, \boldsymbol{s}_2, \ldots, \boldsymbol{s}_C\}$, the core representation $\boldsymbol{o}$ is a vector generated by an arbitrary function $f$ with the following form:

$$\boldsymbol{o} = f(\boldsymbol{s}_1, \boldsymbol{s}_2, \ldots, \boldsymbol{s}_C)$$

The core representation encodes the global information across all the channels. To obtain such representation, we employ the following form, which is inspired by the Kolmogorov-Arnold representation theorem [21] and DeepSets [46]:

$$\boldsymbol{o}_i = \text{Stoch\_Pool}(\text{MLP}_1(\boldsymbol{S}_{i-1})) \tag{3}$$

where $\text{MLP}_1 : \mathbb{R}^d \mapsto \mathbb{R}^{d'}$ is a projection that projects the series representation from the series hidden dimension $d$ to the core dimension $d'$, composing two layers with hidden dimension $d$ and GELU [13] activation. $\text{Stoch\_Pool}$ is the stochastic pooling [47] that get the core representation $\boldsymbol{o} \in \mathbb{R}^{d'}$ by aggregating representations of $C$ series. Stochastic pooling combines the advantages of mean and max pooling. The details of computing the core representation can be found in Appendix B.2. Next, we fuse the representations of the core and all the series:

$$\boldsymbol{F}_i = \text{Repeat\_Concat}(\boldsymbol{S}_{i-1}, \boldsymbol{o}_i) \tag{4}$$

$$\boldsymbol{S}_i = \text{MLP}_2(\boldsymbol{F}_i) + \boldsymbol{S}_{i-1} \tag{5}$$

The $\text{Repeat\_Concat}$ operation concatenate the core representation $\boldsymbol{o}$ to each series representation, and we get the $\boldsymbol{F}_i \in \mathbb{R}^{C \times (d+d')}$. Then another MLP ($\text{MLP}_2 : \mathbb{R}^{d+d'} \mapsto \mathbb{R}^d$) is used to fuse the concatenated presentation and project it back to the hidden dimension $d$, *i.e.*, $\boldsymbol{S}_i \in R^{C \times d}$. Like many deep learning modules, we also add a residual connection from the input to the output [12].

### 3.3 Complexity Analysis

We analyze the complexity of each component of SOFTS step by step concerning window length $L$, number of channels $C$, model dimension $d$, and forecasting horizon $H$. The complexity of the reversible instance normalization and series embedding is $O(CL)$ and $O(CLd)$ respectively. In STAR, assuming $d' = d$, the $\text{MLP}_1$ is a $\mathbb{R}^d \mapsto \mathbb{R}^d$ mapping with complexity $O(Cd^2)$. $\text{Stoch\_Pool}$ computes the softmax along the channel dimension, with complexity $O(Cd)$. The $\text{MLP}_2$ on the concatenated embedding has the complexity $O(Cd^2)$. The complexity of the predictor is $O(CdH)$. In all, the complexity of the encoding part is $O(CLd + Cd^2 + CdH)$, which is linear to $C$, $L$, and $H$. Ignoring the model dimension $d$, which is a constant in the algorithm and irrelevant to the problem, we compare the complexity of several popular forecasters in Table 1.

Table 1: Complexity comparison between popular time series forecasters concerning window length $L$, number of channels $C$ and forecasting horizon $H$. Our method achieves only linear complexity.

|  | SOFTS (ours) | iTransformer | PatchTST | Transformer |
|---|---|---|---|---|
| Complexity | $\boldsymbol{O(CL + CH)}$ | $O(C^2 + CL + CH)$ | $O(CL^2 + CH)$ | $O(CL + L^2 + HL + CH)$ |

## 4 Experiments

**Datasets.** To thoroughly evaluate the performance of our proposed SOFTS, we conduct extensive experiments on 6 widely used, real-world datasets including ETT (4 subsets), Traffic, Electricity, Weather [51, 37], Solar-Energy [22] and PEMS (4 subsets) [25]. Detailed descriptions of the datasets can be found in Appendix A.

### 4.1 Forecasting Results

**Compared methods.** We extensively compare the recent Linear-based or MLP-based methods, including DLinear [48], TSMixer [7], TiDE [6]. We also consider Transformer-based methods including FEDformer [52], Stationary [26], PatchTST [29], Crossformer [50], iTransformer [27] and CNN-based methods including SCINet [25], TimesNet [38].

**Forecasting benchmarks.** The long-term forecasting benchmarks follow the setting in Informer [51] and SCINet [25]. The lookback window length ($L$) is set to 96 for all datasets. We set the prediction horizon ($H$) to $\{12, 24, 48, 96\}$ for PEMS and $\{96, 192, 336, 720\}$ for others. Performance comparison among different methods is conducted based on two primary evaluation metrics: Mean Squared Error (MSE) and Mean Absolute Error (MAE). The results of PatchTST and TSMixer are reproduced for the ablation study and other results are taken from iTransformer [27].

**Implementation details.** We use the ADAM optimizer [19] with an initial learning rate of $3 \times 10^{-4}$. This rate is modulated by a cosine learning rate scheduler. The mean squared error (MSE) loss function is utilized for model optimization. We explore the number of STAR blocks $N$ within the set $\{1, 2, 3, 4\}$, and the dimension of series $d$ within $\{128, 256, 512\}$. Additionally, the dimension of the core representation $d'$ varies among $\{64, 128, 256, 512\}$. Other detailed implementations are described in Appendix B.3.

**Main results.** As shown in Table 2, SOFTS has provided the best or second predictive outcomes in all 6 datasets on average. Moreover, when compared to previous state-of-the-art methods, SOFTS has demonstrated significant advancements. For instance, on the Traffic dataset, SOFTS improved the average MSE error from 0.428 to 0.409, representing a notable reduction of about 4.4%. On the PEMS07 dataset, SOFTS achieves a substantial relative decrease of 13.9% in average MSE error, from 0.101 to 0.087. These significant improvements indicate that the SOFTS model possesses robust performance and broad applicability in multivariate time series forecasting tasks, especially in tasks with a large number of channels, such as the Traffic dataset, which includes 862 channels, and the PEMS dataset, with a varying range from 170 to 883 channels.

Table 2: Multivariate forecasting results with horizon $H \in \{12, 24, 48, 96\}$ for PEMS and $H \in \{96, 192, 336, 720\}$ for others and fixed lookback window length $L = 96$. Results are averaged from all prediction horizons. Full results are listed in Table 6.

| Models | SOFTS (ours) | | iTransformer | | PatchTST | | TSMixer | | Crossformer | | TiDE | | TimesNet | | DLinear | | SCINet | | FEDformer | | Stationary | |
|---|---|---|---|---|---|---|---|---|---|---|---|---|---|---|---|---|---|---|---|---|---|---|
| Metric | MSE | MAE | MSE | MAE | MSE | MAE | MSE | MAE | MSE | MAE | MSE | MAE | MSE | MAE | MSE | MAE | MSE | MAE | MSE | MAE | MSE | MAE |
| ECL | **0.174** | **0.264** | 0.178 | 0.270 | 0.189 | 0.276 | 0.186 | 0.287 | 0.244 | 0.334 | 0.251 | 0.344 | 0.192 | 0.295 | 0.212 | 0.300 | 0.268 | 0.365 | 0.214 | 0.327 | 0.193 | 0.296 |
| Traffic | **0.409** | **0.267** | 0.428 | 0.282 | 0.454 | 0.286 | 0.522 | 0.357 | 0.550 | 0.304 | 0.760 | 0.473 | 0.620 | 0.336 | 0.625 | 0.383 | 0.804 | 0.509 | 0.610 | 0.376 | 0.624 | 0.340 |
| Weather | **0.255** | **0.278** | 0.258 | **0.278** | 0.256 | 0.279 | 0.256 | 0.279 | 0.259 | 0.315 | 0.271 | 0.320 | 0.259 | 0.287 | 0.265 | 0.317 | 0.292 | 0.363 | 0.309 | 0.360 | 0.288 | 0.314 |
| Solar-Energy | **0.229** | **0.256** | 0.233 | 0.262 | 0.236 | 0.266 | 0.260 | 0.297 | 0.641 | 0.639 | 0.347 | 0.417 | 0.301 | 0.319 | 0.330 | 0.401 | 0.282 | 0.375 | 0.291 | 0.381 | 0.261 | 0.381 |
| ETTm1 | **0.393** | **0.403** | 0.407 | 0.410 | 0.396 | 0.406 | 0.398 | 0.407 | 0.513 | 0.496 | 0.419 | 0.419 | 0.400 | 0.406 | 0.403 | 0.407 | 0.485 | 0.481 | 0.448 | 0.452 | 0.481 | 0.456 |
| ETTm2 | **0.287** | **0.330** | 0.288 | 0.332 | 0.287 | **0.330** | 0.289 | 0.333 | 0.757 | 0.610 | 0.358 | 0.404 | 0.291 | 0.333 | 0.350 | 0.401 | 0.571 | 0.537 | 0.305 | 0.349 | 0.306 | 0.347 |
| ETTh1 | 0.449 | **0.442** | 0.454 | 0.447 | 0.453 | 0.446 | 0.463 | 0.452 | 0.529 | 0.522 | 0.541 | 0.507 | 0.458 | 0.450 | 0.456 | 0.452 | 0.747 | 0.647 | **0.440** | 0.460 | 0.570 | 0.537 |
| ETTh2 | **0.373** | **0.400** | 0.383 | 0.407 | 0.385 | 0.410 | 0.401 | 0.417 | 0.942 | 0.684 | 0.611 | 0.550 | 0.414 | 0.427 | 0.559 | 0.515 | 0.954 | 0.723 | 0.437 | 0.449 | 0.526 | 0.516 |
| PEMS03 | **0.104** | **0.210** | 0.113 | 0.221 | 0.137 | 0.240 | 0.119 | 0.233 | 0.169 | 0.281 | 0.326 | 0.419 | 0.147 | 0.248 | 0.278 | 0.375 | 0.114 | 0.224 | 0.213 | 0.327 | 0.147 | 0.249 |
| PEMS04 | 0.102 | 0.208 | 0.111 | 0.221 | 0.145 | 0.249 | 0.103 | 0.215 | 0.209 | 0.314 | 0.353 | 0.437 | 0.129 | 0.241 | 0.295 | 0.388 | **0.092** | **0.202** | 0.231 | 0.337 | 0.127 | 0.240 |
| PEMS07 | **0.087** | **0.184** | 0.101 | 0.204 | 0.144 | 0.233 | 0.112 | 0.217 | 0.235 | 0.315 | 0.380 | 0.440 | 0.124 | 0.225 | 0.329 | 0.395 | 0.119 | 0.234 | 0.165 | 0.283 | 0.127 | 0.230 |
| PEMS08 | **0.138** | **0.219** | 0.150 | 0.226 | 0.200 | 0.275 | 0.165 | 0.261 | 0.268 | 0.307 | 0.441 | 0.464 | 0.193 | 0.271 | 0.379 | 0.416 | 0.158 | 0.244 | 0.286 | 0.358 | 0.201 | 0.276 |

**Model efficiency.** Our SOFTS model demonstrates efficient performance with minimal memory and time consumption. Figure 3b illustrates the memory and time usage across different models on the Traffic dataset, with lookback window $L = 96$, horizon $H = 720$, and batch size $4$. Despite their low resource usage, Linear-based or MLP-based models such as DLinear and TSMixer perform poorly with a large number of channels. Figure 3a explores the memory requirements of the three best-performing models from Figure 3b. This figure reveals that the memory usage of both PatchTST and iTransformer escalates significantly with an increase in channels. In contrast, our SOFTS model maintains efficient operation, with its complexity scaling linearly with the number of channels, effectively handling large channel counts.

## 4.2 Ablation Study

In this section, the prediction horizon ($H$) is set to $\{12, 24, 48, 96\}$ for PEMS and $\{96, 192, 336, 720\}$ for others. All the results are averaged on four horizons. If not especially concerned, the lookback window length ($L$) is set to 96 as default.

**Comparison of different pooling methods.** The comparison of different pooling methods in STAR is shown in Table 3. The term "w/o STAR" refers to a scenario where an MLP is utilized with the Channel Independent (CI) strategy, without the use of STAR. **Mean** pooling computes the average value of all the series representations. **Max** pooling selects the maximum value of each hidden feature among all the channels. **Weighted** average learns the weight for each channel. **Stochastic** pooling applies random selection during training and weighted average during testing according to the feature value. The result reveals that incorporating STAR into the model leads to a consistent enhancement

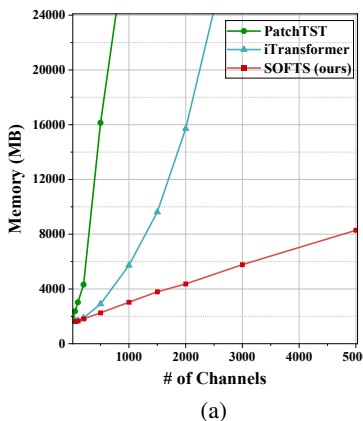
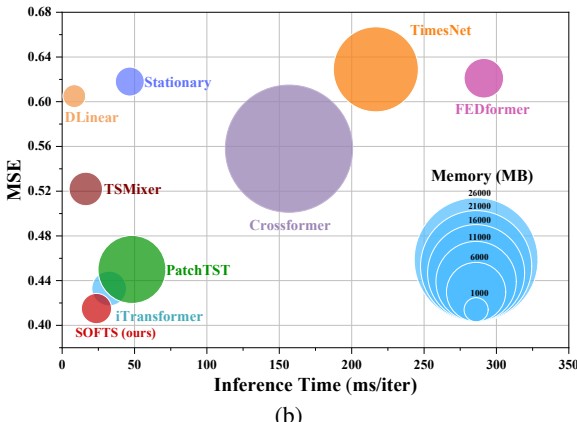

| (a) | (b) |

Figure 3: Memory and time consumption of different models. In Figure 3a, we set the lookback window $L = 96$, horizon $H = 720$, and batch size to 16 in a synthetic dataset we conduct. In Figure 3b, we set the lookback window $L = 96$, horizon $H = 720$, and batch size to 4 in Traffic dataset. Figure 3a reveals that SOFTS model scales to large number of channels more effectively than Transformer-based models. Figure 3b shows that previous Linear-based or MLP-based models such as DLinear and TSMixer perform poorly with a large number of channels. While SOFTS model demonstrates efficient performance with minimal memory and time consumption.

in performance across all pooling methods. Additionally, stochastic pooling deserves attention as it outperforms the other methods across nearly all the datasets.

Table 3: Comparison of the effect of different pooling methods. The term "w/o STAR" refers to a scenario where an MLP is utilized with the Channel Independent (CI) strategy, without the use of STAR. The result reveals that incorporating STAR into the model leads to a consistent enhancement in performance across all pooling methods. Apart from that, stochastic pooling performs better than mean and max pooling. Full results can be found in Table 7.

| Pooling Method | ECL | | Traffic | | Weather | | Solar | | ETTh2 | | PEMS04 | |
|---|---|---|---|---|---|---|---|---|---|---|---|---|
| | MSE | MAE | MSE | MAE | MSE | MAE | MSE | MAE | MSE | MAE | MSE | MAE |
| w/o STAR | 0.187 | 0.273 | 0.442 | 0.281 | 0.261 | 0.281 | 0.247 | 0.272 | 0.381 | 0.406 | 0.143 | 0.245 |
| Mean | **0.174** | 0.266 | 0.420 | 0.277 | 0.261 | 0.281 | 0.234 | 0.262 | 0.379 | 0.404 | 0.106 | 0.212 |
| Max | 0.180 | 0.270 | **0.406** | 0.271 | 0.259 | 0.280 | 0.246 | 0.269 | 0.379 | 0.401 | 0.116 | 0.223 |
| Weighted | 0.184 | 0.275 | 0.440 | 0.292 | 0.263 | 0.284 | 0.264 | 0.280 | 0.379 | 0.403 | 0.109 | 0.218 |
| Stochastic | **0.174** | **0.264** | 0.409 | **0.267** | **0.255** | **0.278** | **0.229** | **0.256** | **0.373** | **0.400** | **0.102** | **0.208** |

**Universality of STAR.** The STar Aggregate-Redistribute (STAR) module is an embedding adaptation function [41, 43] that is replaceable to arbitrary transformer-based methods that use the attention mechanism. In this paragraph, we test the effectiveness of STAR on different existing transformer-based forecasters, such as PatchTST [29] and Crossformer [50]. Note that our method can be regarded as replacing the channel attention in iTransformer [27]. Here we involve substituting the time attention in PatchTST with STAR and incrementally replacing both the time and channel attention in Crossformer with STAR. The results, as presented in Table 4, demonstrate that replacing attention with STAR, which deserves less computational resources, could maintain and even improve the models' performance in several datasets.

**Influence of lookback window length.** Common sense suggests that a longer lookback window should improve forecast accuracy. However, incorporating too many features can lead to a curse

Table 4: The performance of STAR in different models. The attention replaced by STAR here are the time attention in PatchTST, the channel attention in iTransformer, and both the time attention and channel attention in modified Crossformer. The results demonstrate that replacing attention with STAR, which requires less computational resources, could maintain and even improve the models' performance in several datasets. [†]: The Crossformer used here is a modified version that replaces the decoder with a flattened head like what PatchTST does. Full results can be found in Table 8.

| Model | Component | ECL | | Traffic | | Weather | | PEMS03 | | PEMS04 | | PEMS07 | |
|---|---|---|---|---|---|---|---|---|---|---|---|---|---|
| | | MSE | MAE | MSE | MAE | MSE | MAE | MSE | MAE | MSE | MAE | MSE | MAE |
| PatchTST | Attention | 0.189 | 0.276 | 0.454 | 0.286 | 0.256 | 0.279 | 0.137 | 0.240 | 0.145 | 0.249 | 0.144 | 0.233 |
| | STAR | **0.185** | **0.272** | **0.448** | **0.279** | **0.252** | **0.277** | **0.134** | **0.233** | **0.136** | **0.238** | **0.137** | **0.225** |
| Crossformer[†] | Attention | 0.202 | 0.301 | **0.546** | 0.297 | 0.254 | 0.310 | 0.100 | 0.208 | 0.090 | 0.198 | 0.084 | 0.181 |
| | STAR | **0.198** | **0.292** | 0.549 | **0.292** | **0.252** | **0.305** | **0.100** | **0.204** | **0.087** | **0.194** | **0.080** | **0.175** |
| iTransformer | Attention | 0.178 | 0.270 | 0.428 | 0.282 | 0.258 | 0.278 | 0.113 | 0.221 | 0.111 | 0.221 | 0.101 | 0.204 |
| | STAR | **0.174** | **0.264** | **0.409** | **0.267** | **0.255** | **0.278** | **0.104** | **0.210** | **0.102** | **0.208** | **0.087** | **0.184** |

of dimensionality, potentially compromising the model's forecasting effectiveness. We explore how varying the lengths of these lookback windows impacts the forecasting performance for time horizons from 48 to 336 in all datasets. As shown in Figure 4, SOFTS could consistently improve its performance by effectively utilizing the enhanced data available from an extended lookback window. Also, SOFTS performs consistently better than other models under different lookback window lengths, especially in shorter cases.

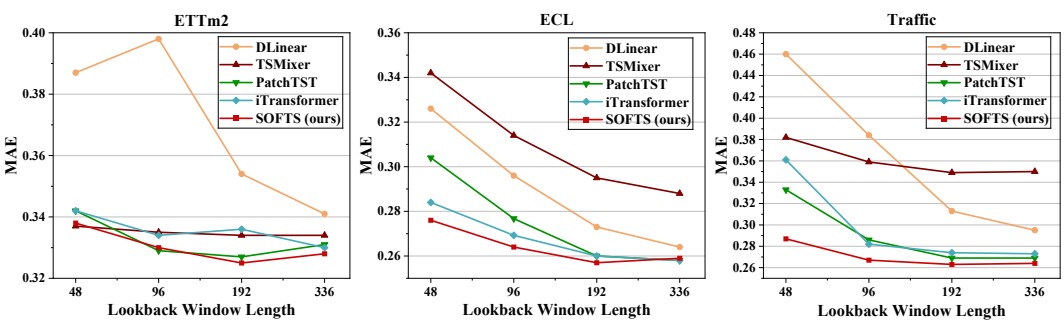

Figure 4: Influence of lookback window length $L$. SOFTS performs consistently better than other models under different lookback window lengths, especially in shorter cases.

**Hyperparameter sensitivity analysis.** We investigate the impact of several key hyperparameters on our model's performance: the hidden dimension of the model, denoted as $d$, the hidden dimension of the core, represented by $d'$, and the number of encoder layers, $N$. Analysis of Figure 5 indicates that complex traffic datasets (such as Traffic and PEMS) require larger hidden dimensions and more encoding layers to handle their intricacies effectively. Moreover, variations in $d'$ have a minimal influence on the model's overall performance.

**Series embedding adaptation of STAR.** The STAR module adapts the series embeddings by extracting the interaction between channels. To give an intuition of the functionality of STAR, we visualize the series embeddings before and after being adjusted by STAR. The multivariate series is selected from the test set of Traffic with look back window 96 and number of channels 862. Figure 6 shows the series embeddings visualized by T-SNE before and after the first STAR module. Among the 862 channels, there are 2 channels embedded far away from the other channels. These two channels can be seen as anomalies, marked as (⋆) in the figure. Without STAR, *i.e.*, using only the channel independent strategy, the prediction on the series can only achieve 0.414 MSE. After being adjusted by STAR, the abnormal channels can be clustered towards normal channels by exchanging channel

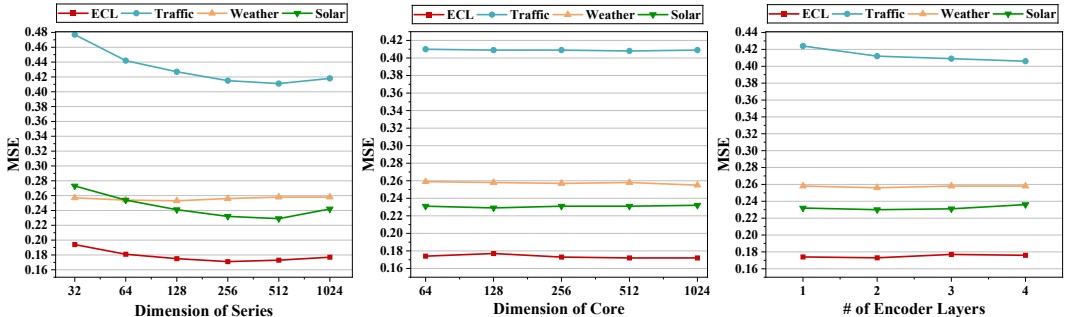

Figure 5: Impact of several key hyperparameters: the hidden dimension of the model, denoted as $d$, the hidden dimension of the core, represented by $d'$, and the number of encoder layers, $N$. Full results can be seen in Appendix C.5.

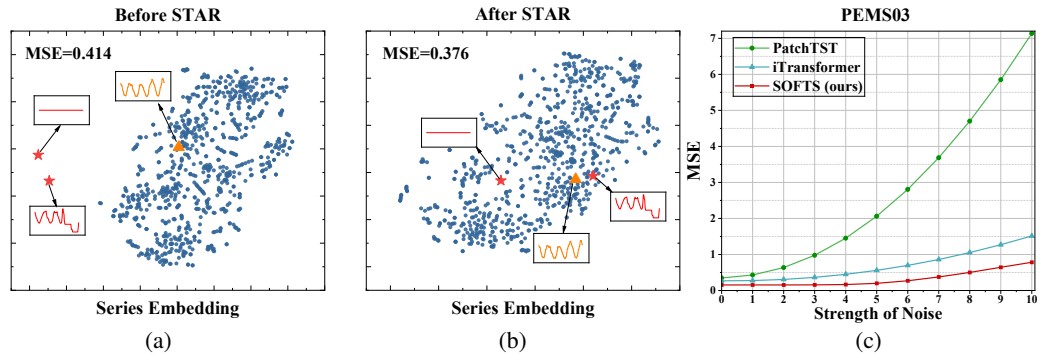

(a)  (b)  (c)

Figure 6: Figure 6a 6b: T-SNE of the series embeddings on the Traffic dataset. 6a: the series embeddings before STAR. Two abnormal channels ($\star$) are located far from the other channels. Forecasting on the embeddings achieves 0.414 MSE. 6b: series embeddings after being adjusted by STAR. The two channels are clustered towards normal channels ($\triangle$) by exchanging channel information. Adapted series embeddings improve forecasting performance to 0.376. Figure 6c: Impact of noise on one channel. Our method is more robust against channel noise than other methods.

information. An example of the normal channels is marked as ($\triangle$). Predictions on the adapted series embeddings can improve the performance to 0.376, a **9%** improvement.

**Impact of channel noise.**  As previously mentioned, SOFTS can cluster abnormal channels towards normal channels by exchanging channel information. To test the impact of an abnormal channel on the performance of three models—SOFTS, PatchTST, and iTransformer—we select one channel from the PEMS03 dataset and add Gaussian noise with a mean of 0 and a standard deviation representing the strength of the noise. The lookback window and horizon are set to 96 for this experiment. In Figure 6c, we observe that the MSE of PatchTST increases sharply as the strength of the noise grows. In contrast, SOFTS and iTransformer can better handle the noise. This indicates that suitable channel interaction can improve the robustness against noise in one channel using information from the normal channels. Moreover, SOFTS demonstrates superior noise handling compared to iTransformer. This suggests that while the abnormal channel can affect the model's judgment of normal channels, our STAR module can mitigate the negative impact more effectively by utilizing core representation instead of building relationships between every pair of channels.

## 5  Conclusion

Although channel independence has been found an effective strategy to improve robustness for multivariate time series forecasting, channel correlation is important information to be utilized

for further improvement. The previous methods faced a dilemma between model complexity and performance in extracting the correlation. In this paper, we solve the dilemma by introducing the Series-cOre Fused Time Series forecaster (SOFTS) which achieves state-of-the-art performance with low complexity, along with a novel STar Aggregate-Redistribute (STAR) module to efficiently capture the channel correlation.

Our paper explores the way of building a scalable multivariate time series forecaster while maintaining equal or even better performance than the state-of-the-art methods, which we think may pave the way to forecasting on datasets of more larger scale under resource constraints [53].

## Acknowledgments

This research was supported by National Science and Technology Major Project (2022ZD0114805), NSFC (61773198, 62376118,61921006), Collaborative Innovation Center of Novel Software Technology and Industrialization, CCF-Tencent Rhino-Bird Open Research Fund (RAGR20240101).

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

# A Datasets Description

We detail the description plus the link to download them here:

1. **ETT (Electricity Transformer Temperature)** [51] [3] comprises two hourly-level datasets (ETTh) and two 15-minute-level datasets (ETTm). Each dataset contains seven oil and load features of electricity transformers from July 2016 to July 2018.

2. **Traffic**[4] describes the road occupancy rates. It contains the hourly data recorded by the sensors of San Francisco freeways from 2015 to 2016.

3. **Electricity**[5] collects the hourly electricity consumption of 321 clients from 2012 to 2014.

4. **Weather** includes 21 indicators of weather, such as air temperature, and humidity. Its data is recorded every 10 min for 2020 in Germany.

5. **Solar-Energy** [22] records the solar power production of 137 PV plants in 2006, which is sampled every 10 minutes.

6. **PEMS** [6] contains public traffic network data in California collected by 5-minute windows.

Other details of these datasets have been concluded in Table 5.

Table 5: Detailed dataset descriptions. *Channels* denotes the number of channels in each dataset. *Dataset Split* denotes the total number of time points in (Train, Validation, Test) split respectively. *Prediction Length* denotes the future time points to be predicted and four prediction settings are included in each dataset. *Granularity* denotes the sampling interval of time points.

| Dataset | Channels | Prediction Length | Dataset Split | Granularity | Domain |
|---|---|---|---|---|---|
| ETTh1, ETTh2 | 7 | {96, 192, 336, 720} | (8640, 2880, 2880) | Hourly | Electricity |
| ETTm1, ETTm2 | 7 | {96, 192, 336, 720} | (34560, 11520, 11520) | 15min | Electricity |
| Weather | 21 | {96, 192, 336, 720} | (36887, 5270, 10539) | 10min | Weather |
| ECL | 321 | {96, 192, 336, 720} | (18412, 2632, 5260) | Hourly | Electricity |
| Traffic | 862 | {96, 192, 336, 720} | (12280, 1756, 3508) | Hourly | Transportation |
| Solar-Energy | 137 | {96, 192, 336, 720} | (36792, 5256, 10512) | 10min | Energy |
| PEMS03 | 358 | {12, 24, 48, 96} | (15724, 5242, 5242) | 5min | Transportation |
| PEMS04 | 307 | {12, 24, 48, 96} | (10195, 3398, 3399) | 5min | Transportation |
| PEMS07 | 883 | {12, 24, 48, 96} | (16934, 5645, 5645) | 5min | Transportation |
| PEMS08 | 170 | {12, 24, 48, 96} | (10713, 3571, 3572) | 5min | Transportation |

# B Implement Details

## B.1 Overall architecture of SOFTS

The overall architecture of SOFTS is detailed in Algorithm 1. Initially, a linear layer is employed to obtain the embedding for each series (Lines 1-2). Subsequently, several encoder layers are applied. Within each encoder layer, the core representation is first derived by applying an MLP to the series embeddings and pooling them (Line 4). This core representation is then concatenated with each series (Line 5), and another MLP is used to fuse them (Line 6). After passing through multiple encoder layers, a final linear layer projects the series embeddings to the predicted series (Line 8).

---

[3] https://github.com/zhouhaoyi/ETDataset
[4] http://pems.dot.ca.gov
[5] https://archive.ics.uci.edu/ml/datasets/ElectricityLoadDiagrams20112014
[6] https://pems.dot.ca.gov/

---

**Algorithm 1** Series-cOre Fused Time Series forecaster (SOFTS) applied to multivariate time series.

---

**Require:** Look back window $\boldsymbol{X} \in \mathbb{R}^{L \times C}$;
 1: $\boldsymbol{X} = \boldsymbol{X}.\text{transpose}$                                                        ▷ $\boldsymbol{X} \in \mathbb{R}^{C \times L}$
 2: $\boldsymbol{S}_0 = \text{Linear}(\boldsymbol{X})$                                        ▷ Get series embedding, $\boldsymbol{S}_0 \in \mathbb{R}^{C \times d}$
 3: **for** $l = 1 \ldots L$ **do**
 4:     $\boldsymbol{o}_i = \text{Stoch\_Pool}(\text{MLP}(\boldsymbol{S}_{i-1}))$                     ▷ Get core representation, $\boldsymbol{o}_i \in \mathbb{R}^{d'}$
 5:     $\boldsymbol{F}_i = \text{Repeat\_Concat}(\boldsymbol{S}_{i-1}, \boldsymbol{o}_i)$                              ▷ $\boldsymbol{F}_i \in \mathbb{R}^{C \times (d+d')}$
 6:     $\boldsymbol{S}_i = \text{MLP}(\boldsymbol{F}_i) + \boldsymbol{S}_{i-1}$                        ▷ Fuse series and core, $\boldsymbol{S}_i \in \mathbb{R}^{C \times d}$
 7: **end for**
 8: $\hat{\boldsymbol{Y}} = \text{Linear}(\boldsymbol{S}_L)$                 ▷ Project series embedding to predicted series, $\hat{\boldsymbol{Y}} \in \mathbb{R}^{C \times H}$
 9: $\hat{\boldsymbol{Y}} = \hat{\boldsymbol{Y}}.\text{transpose}$                                                      ▷ $\hat{\boldsymbol{Y}} \in \mathbb{R}^{H \times C}$
10: **return** $\hat{\boldsymbol{Y}}$

---

## B.2 Details of Core Representation Computation

**Core representation.**    Recall that the core representation for the multivariate time series is defined in Definition 3.1 with the following form:

$$\boldsymbol{o} = f(\boldsymbol{s}_1, \boldsymbol{s}_2, \ldots, \boldsymbol{s}_C)$$

To obtain the representation, we draw inspiration from the two theorems:

**Theorem B.1** (Kolmogorov-Arnold representation [21]). *Let $f : [0,1]^M \to \mathbb{R}$ be an arbitrary multivariate continuous function iff it has the representation*

$$f(x_1, \ldots, x_M) = \rho \left( \sum_{m=1}^{M} \lambda_m \phi(x_m) \right)$$

*with continuous outer and inner functions $\rho : \mathbb{R}^{2M+1} \to \mathbb{R}$ and $\phi : \mathbb{R} \to \mathbb{R}^{2M+1}$. The inner function $\phi$ is independent of the function $f$.*

**Theorem B.2** (DeepSets [46]). *Assume the elements are from a compact set in $\mathbb{R}^d$, i.e. possibly uncountable, and the set size is fixed to $M$. Then any continuous function operating on a set $X$, i.e. $f : \mathbb{R}^{d \times M} \to \mathbb{R}$ which is permutation invariant to the elements in $X$ can be approximated arbitrarily close in the form of*

$$\rho \left( \sum_{x \in X} \phi(x) \right),$$

*for suitable transformations $\phi$ and $\rho$.*

The two formulations are very similar, except for the dependence of inner transformation on the coordinate through $\lambda_m$. The existence of $\lambda$ determines whether the formulation is permutation invariant or not. *In this paper, we find in Table 4 that the permutation invariant expression (Theorem B.2) performs much better than the permutation variant one Theorem B.1.* This may be attributed to the characteristics of channel series being enough to induce the index of each channel (coordinate). Introducing extra parameters specific to each channel may enhance the dependency channel coordinate and reduce the dependence on the history, therefore causing low robustness when encountering unknown series. Consequently, we utilize DeepSets form to express the core representation:

$$\boldsymbol{o} = \rho \left( \sum_{\boldsymbol{s} \in \boldsymbol{S}} \phi(\boldsymbol{s}) \right).$$

We propose two modifications to the expression:

1. We generalize the mean pooling over the inner transformation by arbitrary pooling methods.

2. We remove the outer transformation $\rho$ because it is redundant with the MLP during the fusion process.

For 1., we tested several common pooling methods and found that the mean pooling and max pooling outperform each other in different cases. Stochastic pooling (described in the following paragraph) can achieve the best results in averaged cases (Table 3). So, the core is computed as Equation (3).

**Stochastic pooling.** Stochastic pooling is a pooling method that combines the characteristics of max pooling and mean pooling [47]. In stochastic pooling, the pooled map response is selected by sampling from a multinomial distribution formed from the activations of each pooling region. Specifically, we first calculate the probabilities $p$ for each dimension $j$ by normalizing the softmax activations within the dimension:

$$p_{ij} = \frac{e^{A_{ij}}}{\sum_{k=1}^{C} e^{A_{kj}}} \tag{6}$$

During training, we sample from the multinomial distribution based on $p$ to pick a channel $c$ within the dimension $j$. The pooled result is then simply $A_{cj}$:

$$\boldsymbol{o}_j = A_{cj} \text{ where } c \sim P(p_{1j}, p_{2j}, ..., p_{Cj}) \tag{7}$$

At test time, we use a probabilistic form of averaging:

$$\boldsymbol{o}_j = \sum_{i=1}^{C} p_{ij} A_{ij} \tag{8}$$

This approach allows for a more robust and statistically grounded pooling mechanism, which can enhance the generalization capabilities of the model across different data scenarios.

### B.3 Experiment Details

All the experiments are conducted on a single NVIDIA GeForce RTX 3090 with 24G VRAM. The mean squared error (MSE) loss function is utilized for model optimization. Performance comparison among different methods is conducted based on two primary evaluation metrics: Mean Squared Error (MSE) and Mean Absolute Error (MAE). We use the ADAM optimizer [19] with an initial learning rate of $3 \times 10^{-4}$. This rate is modulated by a cosine learning rate scheduler. We explore the number of STAR blocks $N$ within the set $\{1, 2, 3, 4\}$, and the dimension of series $d$ within $\{128, 256, 512\}$. Additionally, the dimension of the core representation $d'$ is searched in $\{64, 128, 256, 512\}$, with the constraint that $d'$ does not exceed $d$.

**Mean Squared Error (MSE)**:

$$\text{MSE} = \frac{1}{H} \sum_{i=1}^{H} (\mathbf{Y}_i - \hat{\mathbf{Y}}_i)^2 \tag{9}$$

**Mean Absolute Error (MAE)**:

$$\text{MAE} = \frac{1}{H} \sum_{i=1}^{H} |\mathbf{Y}_i - \hat{\mathbf{Y}}_i| \tag{10}$$

where $\mathbf{Y}, \hat{\mathbf{Y}} \in \mathbb{R}^{H \times C}$ are the ground truth and prediction results of the future with $H$ time points and $C$ channels. $\mathbf{Y}_i$ denotes the $i$-th future time point.

## C  Full Results

### C.1  Full Results of Multivariate Forecasting Benchmark

The complete results of our forecasting benchmarks are presented in Table 6. We conducted experiments using six widely utilized real-world datasets and compared our method against ten previous state-of-the-art models. Our approach, SOFTS, demonstrates strong performance across these tests.

### C.2  Full Results of Pooling Method Ablation

The complete results of our pooling method ablation are presented in Table 7. The term "w/o STAR" refers to a scenario where an MLP is utilized with the Channel Independent (CI) strategy, without the use of STAR. **Mean** pooling computes the average value of all the series representations. **Max** pooling selects the maximum value of each hidden feature among all the channels. **Weighted** average learns the weight for each channel. **Stochastic** pooling applies random selection during training and weighted average during testing according to the feature value. The result reveals that incorporating STAR into the model leads to a consistent enhancement in performance across all pooling methods.

Table 6: Multivariate forecasting results with prediction lengths $H \in \{12, 24, 48, 96\}$ for PEMS and $H \in \{96, 192, 336, 720\}$ for others and fixed lookback window length $L = 96$. The results of PatchTST and TSMixer are reproduced for the ablation study and other results are taken from iTransformer [27].

| Models | | SOFTS (ours) | | iTransformer | | PatchTST | | TSMixer | | Crossformer | | TiDE | | TimesNet | | DLinear | | SCINet | | FEDformer | | Stationary | |
|---|---|---|---|---|---|---|---|---|---|---|---|---|---|---|---|---|---|---|---|---|---|---|---|
| Metric | | MSE | MAE | MSE | MAE | MSE | MAE | MSE | MAE | MSE | MAE | MSE | MAE | MSE | MAE | MSE | MAE | MSE | MAE | MSE | MAE | MSE | MAE |
| ETTm1 | 96 | 0.325 | 0.361 | 0.334 | 0.368 | 0.329 | 0.365 | 0.323 | 0.363 | 0.404 | 0.426 | 0.364 | 0.387 | 0.338 | 0.375 | 0.345 | 0.372 | 0.418 | 0.438 | 0.379 | 0.419 | 0.386 | 0.398 |
| | 192 | 0.375 | 0.389 | 0.377 | 0.391 | 0.380 | 0.394 | 0.376 | 0.392 | 0.450 | 0.451 | 0.398 | 0.404 | 0.374 | 0.387 | 0.380 | 0.389 | 0.439 | 0.450 | 0.426 | 0.441 | 0.459 | 0.444 |
| | 336 | 0.405 | 0.412 | 0.426 | 0.420 | 0.400 | 0.410 | 0.407 | 0.413 | 0.532 | 0.515 | 0.428 | 0.425 | 0.410 | 0.411 | 0.413 | 0.413 | 0.490 | 0.485 | 0.445 | 0.459 | 0.495 | 0.464 |
| | 720 | 0.466 | 0.447 | 0.491 | 0.459 | 0.475 | 0.453 | 0.485 | 0.459 | 0.666 | 0.589 | 0.487 | 0.461 | 0.478 | 0.450 | 0.474 | 0.453 | 0.595 | 0.550 | 0.543 | 0.490 | 0.585 | 0.516 |
| | Avg | 0.393 | 0.403 | 0.407 | 0.410 | 0.396 | 0.406 | 0.398 | 0.407 | 0.513 | 0.496 | 0.419 | 0.419 | 0.400 | 0.406 | 0.403 | 0.407 | 0.485 | 0.481 | 0.448 | 0.452 | 0.481 | 0.456 |
| ETTm2 | 96 | 0.180 | 0.261 | 0.180 | 0.264 | 0.184 | 0.264 | 0.182 | 0.266 | 0.287 | 0.366 | 0.207 | 0.305 | 0.187 | 0.267 | 0.193 | 0.292 | 0.286 | 0.377 | 0.203 | 0.287 | 0.192 | 0.274 |
| | 192 | 0.246 | 0.306 | 0.250 | 0.309 | 0.246 | 0.306 | 0.249 | 0.309 | 0.414 | 0.492 | 0.290 | 0.364 | 0.249 | 0.309 | 0.284 | 0.362 | 0.399 | 0.445 | 0.269 | 0.328 | 0.280 | 0.339 |
| | 336 | 0.319 | 0.352 | 0.311 | 0.348 | 0.308 | 0.346 | 0.309 | 0.347 | 0.597 | 0.542 | 0.377 | 0.422 | 0.321 | 0.351 | 0.369 | 0.427 | 0.637 | 0.591 | 0.325 | 0.366 | 0.334 | 0.361 |
| | 720 | 0.405 | 0.401 | 0.412 | 0.407 | 0.409 | 0.402 | 0.416 | 0.408 | 1.730 | 1.042 | 0.558 | 0.524 | 0.408 | 0.403 | 0.554 | 0.522 | 0.960 | 0.735 | 0.421 | 0.415 | 0.417 | 0.413 |
| | Avg | 0.287 | 0.330 | 0.288 | 0.332 | 0.287 | 0.330 | 0.289 | 0.333 | 0.757 | 0.610 | 0.358 | 0.404 | 0.291 | 0.333 | 0.350 | 0.401 | 0.571 | 0.537 | 0.305 | 0.349 | 0.306 | 0.347 |
| ETTh1 | 96 | 0.381 | 0.399 | 0.386 | 0.405 | 0.394 | 0.406 | 0.401 | 0.412 | 0.423 | 0.448 | 0.479 | 0.464 | 0.384 | 0.402 | 0.386 | 0.400 | 0.654 | 0.599 | 0.376 | 0.419 | 0.513 | 0.491 |
| | 192 | 0.435 | 0.431 | 0.441 | 0.436 | 0.440 | 0.435 | 0.452 | 0.442 | 0.471 | 0.474 | 0.525 | 0.492 | 0.436 | 0.429 | 0.437 | 0.432 | 0.719 | 0.631 | 0.420 | 0.448 | 0.534 | 0.504 |
| | 336 | 0.480 | 0.452 | 0.487 | 0.458 | 0.491 | 0.462 | 0.492 | 0.463 | 0.570 | 0.546 | 0.565 | 0.515 | 0.491 | 0.469 | 0.481 | 0.459 | 0.778 | 0.659 | 0.459 | 0.465 | 0.588 | 0.535 |
| | 720 | 0.499 | 0.488 | 0.503 | 0.491 | 0.487 | 0.479 | 0.507 | 0.490 | 0.653 | 0.621 | 0.594 | 0.558 | 0.521 | 0.500 | 0.519 | 0.516 | 0.836 | 0.699 | 0.506 | 0.507 | 0.643 | 0.616 |
| | Avg | 0.449 | 0.442 | 0.454 | 0.447 | 0.453 | 0.446 | 0.463 | 0.452 | 0.529 | 0.522 | 0.541 | 0.507 | 0.458 | 0.450 | 0.456 | 0.452 | 0.747 | 0.647 | 0.440 | 0.460 | 0.570 | 0.537 |
| ETTh2 | 96 | 0.297 | 0.347 | 0.297 | 0.349 | 0.288 | 0.340 | 0.319 | 0.361 | 0.745 | 0.584 | 0.400 | 0.440 | 0.340 | 0.374 | 0.333 | 0.387 | 0.707 | 0.621 | 0.358 | 0.397 | 0.476 | 0.458 |
| | 192 | 0.373 | 0.394 | 0.380 | 0.400 | 0.376 | 0.395 | 0.402 | 0.410 | 0.877 | 0.656 | 0.528 | 0.509 | 0.402 | 0.414 | 0.477 | 0.476 | 0.860 | 0.689 | 0.429 | 0.439 | 0.512 | 0.493 |
| | 336 | 0.410 | 0.426 | 0.428 | 0.432 | 0.440 | 0.451 | 0.444 | 0.446 | 1.043 | 0.731 | 0.643 | 0.571 | 0.452 | 0.452 | 0.594 | 0.541 | 1.000 | 0.744 | 0.496 | 0.487 | 0.552 | 0.551 |
| | 720 | 0.411 | 0.433 | 0.427 | 0.445 | 0.436 | 0.453 | 0.441 | 0.450 | 1.104 | 0.763 | 0.874 | 0.679 | 0.462 | 0.468 | 0.831 | 0.657 | 1.249 | 0.838 | 0.463 | 0.474 | 0.562 | 0.560 |
| | Avg | 0.373 | 0.400 | 0.383 | 0.407 | 0.385 | 0.410 | 0.401 | 0.417 | 0.942 | 0.684 | 0.611 | 0.550 | 0.414 | 0.427 | 0.559 | 0.515 | 0.954 | 0.723 | 0.437 | 0.449 | 0.526 | 0.516 |
| ECL | 96 | 0.143 | 0.233 | 0.148 | 0.240 | 0.164 | 0.251 | 0.157 | 0.260 | 0.219 | 0.314 | 0.237 | 0.329 | 0.168 | 0.272 | 0.197 | 0.282 | 0.247 | 0.345 | 0.193 | 0.308 | 0.169 | 0.273 |
| | 192 | 0.158 | 0.248 | 0.162 | 0.253 | 0.173 | 0.262 | 0.173 | 0.274 | 0.231 | 0.322 | 0.236 | 0.330 | 0.184 | 0.289 | 0.196 | 0.285 | 0.257 | 0.355 | 0.201 | 0.315 | 0.182 | 0.286 |
| | 336 | 0.178 | 0.269 | 0.178 | 0.269 | 0.190 | 0.279 | 0.192 | 0.295 | 0.246 | 0.337 | 0.249 | 0.344 | 0.198 | 0.300 | 0.209 | 0.301 | 0.269 | 0.369 | 0.214 | 0.329 | 0.200 | 0.304 |
| | 720 | 0.218 | 0.305 | 0.225 | 0.317 | 0.230 | 0.313 | 0.223 | 0.318 | 0.280 | 0.363 | 0.284 | 0.373 | 0.220 | 0.320 | 0.245 | 0.333 | 0.299 | 0.390 | 0.246 | 0.355 | 0.222 | 0.321 |
| | Avg | 0.174 | 0.264 | 0.178 | 0.270 | 0.189 | 0.276 | 0.186 | 0.287 | 0.244 | 0.334 | 0.251 | 0.344 | 0.192 | 0.295 | 0.212 | 0.300 | 0.268 | 0.365 | 0.214 | 0.327 | 0.193 | 0.296 |
| Traffic | 96 | 0.376 | 0.251 | 0.395 | 0.268 | 0.427 | 0.272 | 0.493 | 0.336 | 0.522 | 0.290 | 0.805 | 0.493 | 0.593 | 0.321 | 0.650 | 0.396 | 0.788 | 0.499 | 0.587 | 0.366 | 0.612 | 0.338 |
| | 192 | 0.398 | 0.261 | 0.417 | 0.276 | 0.454 | 0.289 | 0.497 | 0.351 | 0.530 | 0.293 | 0.756 | 0.474 | 0.617 | 0.336 | 0.598 | 0.370 | 0.789 | 0.505 | 0.604 | 0.373 | 0.613 | 0.340 |
| | 336 | 0.415 | 0.269 | 0.433 | 0.283 | 0.450 | 0.282 | 0.528 | 0.361 | 0.558 | 0.305 | 0.762 | 0.477 | 0.629 | 0.336 | 0.605 | 0.373 | 0.797 | 0.508 | 0.621 | 0.383 | 0.618 | 0.328 |
| | 720 | 0.447 | 0.287 | 0.467 | 0.302 | 0.484 | 0.301 | 0.569 | 0.380 | 0.589 | 0.328 | 0.719 | 0.449 | 0.640 | 0.350 | 0.645 | 0.394 | 0.841 | 0.523 | 0.626 | 0.382 | 0.653 | 0.355 |
| | Avg | 0.409 | 0.267 | 0.428 | 0.282 | 0.454 | 0.286 | 0.522 | 0.357 | 0.550 | 0.304 | 0.760 | 0.473 | 0.620 | 0.336 | 0.625 | 0.383 | 0.804 | 0.509 | 0.610 | 0.376 | 0.624 | 0.340 |
| Weather | 96 | 0.166 | 0.208 | 0.174 | 0.214 | 0.176 | 0.217 | 0.166 | 0.210 | 0.158 | 0.230 | 0.202 | 0.261 | 0.172 | 0.220 | 0.196 | 0.255 | 0.221 | 0.306 | 0.217 | 0.296 | 0.173 | 0.223 |
| | 192 | 0.217 | 0.253 | 0.221 | 0.254 | 0.221 | 0.256 | 0.215 | 0.256 | 0.206 | 0.277 | 0.242 | 0.298 | 0.219 | 0.261 | 0.237 | 0.296 | 0.261 | 0.340 | 0.276 | 0.336 | 0.245 | 0.285 |
| | 336 | 0.282 | 0.300 | 0.278 | 0.296 | 0.275 | 0.296 | 0.287 | 0.300 | 0.272 | 0.335 | 0.287 | 0.335 | 0.280 | 0.306 | 0.283 | 0.335 | 0.309 | 0.378 | 0.339 | 0.380 | 0.321 | 0.338 |
| | 720 | 0.356 | 0.351 | 0.358 | 0.347 | 0.352 | 0.346 | 0.355 | 0.348 | 0.398 | 0.418 | 0.351 | 0.386 | 0.365 | 0.359 | 0.345 | 0.381 | 0.377 | 0.427 | 0.403 | 0.428 | 0.414 | 0.410 |
| | Avg | 0.255 | 0.278 | 0.258 | 0.278 | 0.256 | 0.279 | 0.256 | 0.279 | 0.259 | 0.315 | 0.271 | 0.320 | 0.259 | 0.287 | 0.265 | 0.317 | 0.292 | 0.363 | 0.309 | 0.360 | 0.288 | 0.314 |
| Solar-Energy | 96 | 0.200 | 0.230 | 0.203 | 0.237 | 0.205 | 0.246 | 0.221 | 0.275 | 0.310 | 0.331 | 0.312 | 0.399 | 0.250 | 0.292 | 0.290 | 0.378 | 0.237 | 0.344 | 0.242 | 0.342 | 0.215 | 0.249 |
| | 192 | 0.229 | 0.253 | 0.233 | 0.261 | 0.237 | 0.267 | 0.268 | 0.306 | 0.734 | 0.725 | 0.339 | 0.416 | 0.296 | 0.318 | 0.320 | 0.398 | 0.280 | 0.380 | 0.285 | 0.380 | 0.254 | 0.272 |
| | 336 | 0.243 | 0.269 | 0.248 | 0.273 | 0.250 | 0.276 | 0.272 | 0.294 | 0.750 | 0.735 | 0.368 | 0.430 | 0.319 | 0.330 | 0.353 | 0.415 | 0.304 | 0.389 | 0.282 | 0.376 | 0.290 | 0.296 |
| | 720 | 0.245 | 0.272 | 0.249 | 0.275 | 0.252 | 0.275 | 0.281 | 0.313 | 0.769 | 0.765 | 0.370 | 0.425 | 0.338 | 0.337 | 0.356 | 0.413 | 0.308 | 0.388 | 0.357 | 0.427 | 0.285 | 0.200 |
| | Avg | 0.229 | 0.256 | 0.233 | 0.262 | 0.236 | 0.266 | 0.260 | 0.297 | 0.641 | 0.639 | 0.347 | 0.417 | 0.301 | 0.319 | 0.330 | 0.401 | 0.282 | 0.375 | 0.291 | 0.381 | 0.261 | 0.381 |
| PEMS03 | 12 | 0.064 | 0.165 | 0.071 | 0.174 | 0.073 | 0.178 | 0.075 | 0.186 | 0.090 | 0.203 | 0.178 | 0.305 | 0.085 | 0.192 | 0.122 | 0.243 | 0.066 | 0.172 | 0.126 | 0.251 | 0.081 | 0.188 |
| | 24 | 0.083 | 0.188 | 0.093 | 0.201 | 0.105 | 0.212 | 0.095 | 0.210 | 0.121 | 0.240 | 0.257 | 0.371 | 0.118 | 0.223 | 0.201 | 0.317 | 0.085 | 0.198 | 0.149 | 0.275 | 0.105 | 0.214 |
| | 48 | 0.114 | 0.223 | 0.125 | 0.236 | 0.159 | 0.264 | 0.121 | 0.240 | 0.202 | 0.317 | 0.379 | 0.463 | 0.155 | 0.260 | 0.333 | 0.425 | 0.127 | 0.238 | 0.227 | 0.348 | 0.154 | 0.257 |
| | 96 | 0.156 | 0.264 | 0.164 | 0.275 | 0.210 | 0.305 | 0.184 | 0.295 | 0.262 | 0.367 | 0.490 | 0.539 | 0.228 | 0.317 | 0.457 | 0.515 | 0.178 | 0.287 | 0.348 | 0.434 | 0.247 | 0.336 |
| | Avg | 0.104 | 0.210 | 0.113 | 0.221 | 0.137 | 0.240 | 0.119 | 0.233 | 0.169 | 0.281 | 0.326 | 0.419 | 0.147 | 0.248 | 0.278 | 0.375 | 0.114 | 0.224 | 0.213 | 0.327 | 0.147 | 0.249 |
| PEMS04 | 12 | 0.074 | 0.176 | 0.078 | 0.183 | 0.085 | 0.189 | 0.079 | 0.188 | 0.098 | 0.218 | 0.219 | 0.340 | 0.087 | 0.195 | 0.148 | 0.272 | 0.073 | 0.177 | 0.138 | 0.262 | 0.088 | 0.196 |
| | 24 | 0.088 | 0.194 | 0.095 | 0.205 | 0.115 | 0.222 | 0.089 | 0.201 | 0.131 | 0.256 | 0.292 | 0.398 | 0.103 | 0.215 | 0.224 | 0.340 | 0.084 | 0.193 | 0.177 | 0.293 | 0.104 | 0.216 |
| | 48 | 0.110 | 0.219 | 0.120 | 0.233 | 0.167 | 0.273 | 0.111 | 0.222 | 0.205 | 0.326 | 0.409 | 0.478 | 0.136 | 0.250 | 0.355 | 0.437 | 0.099 | 0.211 | 0.270 | 0.368 | 0.137 | 0.251 |
| | 96 | 0.135 | 0.244 | 0.150 | 0.262 | 0.211 | 0.310 | 0.133 | 0.247 | 0.402 | 0.457 | 0.492 | 0.532 | 0.190 | 0.303 | 0.452 | 0.504 | 0.114 | 0.227 | 0.341 | 0.427 | 0.186 | 0.297 |
| | Avg | 0.102 | 0.208 | 0.111 | 0.221 | 0.145 | 0.249 | 0.103 | 0.215 | 0.209 | 0.314 | 0.353 | 0.437 | 0.129 | 0.241 | 0.295 | 0.388 | 0.092 | 0.202 | 0.231 | 0.337 | 0.127 | 0.240 |
| PEMS07 | 12 | 0.057 | 0.152 | 0.067 | 0.165 | 0.068 | 0.163 | 0.073 | 0.181 | 0.094 | 0.200 | 0.173 | 0.304 | 0.082 | 0.181 | 0.115 | 0.242 | 0.068 | 0.171 | 0.109 | 0.225 | 0.083 | 0.185 |
| | 24 | 0.073 | 0.173 | 0.088 | 0.190 | 0.102 | 0.201 | 0.090 | 0.199 | 0.139 | 0.247 | 0.271 | 0.383 | 0.101 | 0.204 | 0.210 | 0.329 | 0.119 | 0.225 | 0.125 | 0.244 | 0.102 | 0.207 |
| | 48 | 0.096 | 0.195 | 0.110 | 0.215 | 0.170 | 0.261 | 0.124 | 0.231 | 0.311 | 0.369 | 0.446 | 0.495 | 0.134 | 0.238 | 0.398 | 0.458 | 0.149 | 0.237 | 0.165 | 0.288 | 0.136 | 0.240 |
| | 96 | 0.120 | 0.218 | 0.139 | 0.245 | 0.236 | 0.308 | 0.163 | 0.255 | 0.396 | 0.442 | 0.628 | 0.577 | 0.181 | 0.279 | 0.594 | 0.553 | 0.141 | 0.234 | 0.262 | 0.376 | 0.187 | 0.287 |
| | Avg | 0.087 | 0.184 | 0.101 | 0.204 | 0.144 | 0.233 | 0.112 | 0.217 | 0.235 | 0.315 | 0.380 | 0.440 | 0.124 | 0.225 | 0.329 | 0.395 | 0.119 | 0.234 | 0.165 | 0.283 | 0.127 | 0.230 |
| PEMS08 | 12 | 0.074 | 0.171 | 0.079 | 0.182 | 0.098 | 0.205 | 0.083 | 0.189 | 0.165 | 0.214 | 0.227 | 0.343 | 0.112 | 0.212 | 0.154 | 0.276 | 0.087 | 0.184 | 0.173 | 0.273 | 0.109 | 0.207 |
| | 24 | 0.104 | 0.201 | 0.115 | 0.219 | 0.162 | 0.266 | 0.117 | 0.226 | 0.215 | 0.260 | 0.318 | 0.409 | 0.141 | 0.238 | 0.248 | 0.353 | 0.122 | 0.221 | 0.210 | 0.301 | 0.140 | 0.236 |
| | 48 | 0.164 | 0.253 | 0.186 | 0.235 | 0.238 | 0.311 | 0.196 | 0.299 | 0.315 | 0.355 | 0.497 | 0.510 | 0.198 | 0.283 | 0.440 | 0.470 | 0.189 | 0.270 | 0.320 | 0.394 | 0.211 | 0.294 |
| | 96 | 0.211 | 0.253 | 0.221 | 0.267 | 0.303 | 0.318 | 0.266 | 0.331 | 0.377 | 0.397 | 0.721 | 0.592 | 0.320 | 0.351 | 0.674 | 0.565 | 0.236 | 0.300 | 0.442 | 0.465 | 0.345 | 0.367 |
| | Avg | 0.138 | 0.219 | 0.150 | 0.226 | 0.200 | 0.275 | 0.165 | 0.261 | 0.268 | 0.307 | 0.441 | 0.464 | 0.193 | 0.271 | 0.379 | 0.416 | 0.158 | 0.244 | 0.286 | 0.358 | 0.201 | 0.276 |
| 1st Count | | 40 | 47 | 2 | 4 | 6 | 8 | 1 | 0 | 3 | 0 | 0 | 0 | 1 | 2 | 1 | 0 | 5 | 4 | 4 | 0 | 0 | 0 |

Table 7: Comparison of the effect of different pooling methods. The term "w/o STAR" refers to a scenario where an MLP is utilized with the Channel Independent (CI) strategy, without the use of STAR. The result reveals that incorporating STAR into the model leads to a consistent enhancement in performance across all pooling methods. Apart from that, stochastic pooling performs better than mean and max pooling.

| Pooling Method | | w/o STAR | | Mean | | Max | | Weighted | | Stochastic | |
|---|---|---|---|---|---|---|---|---|---|---|---|
| Metric | | MSE | MAE | MSE | MAE | MSE | MAE | MSE | MAE | MSE | MAE |
| ECL | 96 | 0.161 | 0.248 | 0.146 | 0.239 | 0.150 | 0.241 | 0.156 | 0.247 | **0.143** | **0.233** |
| | 192 | 0.171 | 0.259 | 0.166 | 0.258 | 0.165 | 0.256 | 0.173 | 0.264 | **0.158** | **0.248** |
| | 336 | 0.188 | 0.276 | **0.175** | **0.269** | 0.188 | 0.280 | 0.190 | 0.284 | 0.178 | **0.269** |
| | 720 | 0.228 | 0.311 | **0.211** | **0.300** | 0.216 | 0.304 | 0.217 | 0.305 | 0.218 | 0.305 |
| | Avg | 0.187 | 0.273 | **0.174** | 0.266 | 0.180 | 0.270 | 0.184 | 0.275 | **0.174** | **0.264** |
| Traffic | 96 | 0.414 | 0.266 | 0.380 | 0.255 | 0.386 | 0.261 | 0.410 | 0.275 | **0.376** | **0.251** |
| | 192 | 0.428 | 0.272 | 0.406 | 0.268 | **0.397** | 0.267 | 0.434 | 0.288 | 0.398 | **0.261** |
| | 336 | 0.446 | 0.284 | 0.442 | 0.293 | **0.406** | 0.273 | 0.447 | 0.295 | 0.415 | **0.269** |
| | 720 | 0.480 | 0.303 | 0.453 | 0.293 | **0.433** | **0.284** | 0.470 | 0.308 | 0.447 | 0.287 |
| | Avg | 0.442 | 0.281 | 0.420 | 0.277 | **0.406** | 0.271 | 0.440 | 0.292 | 0.409 | **0.267** |
| Weather | 96 | 0.179 | 0.217 | 0.174 | 0.213 | 0.172 | 0.211 | 0.180 | 0.222 | **0.166** | **0.208** |
| | 192 | 0.227 | 0.259 | 0.227 | 0.260 | 0.226 | 0.260 | 0.226 | 0.261 | **0.217** | **0.253** |
| | 336 | 0.281 | 0.299 | 0.281 | 0.299 | **0.280** | **0.298** | 0.284 | 0.302 | 0.282 | 0.300 |
| | 720 | 0.357 | **0.348** | 0.361 | 0.352 | 0.360 | 0.350 | 0.360 | 0.351 | **0.356** | 0.351 |
| | Avg | 0.261 | 0.281 | 0.261 | 0.281 | 0.259 | 0.280 | 0.263 | 0.284 | **0.255** | **0.278** |
| Solar | 96 | 0.215 | 0.250 | 0.202 | 0.238 | 0.206 | 0.243 | 0.219 | 0.260 | **0.200** | **0.230** |
| | 192 | 0.246 | 0.271 | 0.238 | 0.260 | 0.245 | 0.266 | 0.255 | 0.272 | **0.229** | **0.253** |
| | 336 | 0.263 | 0.282 | 0.248 | 0.277 | 0.267 | 0.284 | 0.292 | 0.294 | **0.243** | **0.269** |
| | 720 | 0.263 | 0.283 | 0.247 | **0.271** | 0.265 | 0.284 | 0.290 | 0.293 | **0.245** | 0.272 |
| | Avg | 0.247 | 0.272 | 0.234 | 0.262 | 0.246 | 0.269 | 0.264 | 0.280 | **0.229** | **0.256** |
| ETTh2 | 96 | 0.298 | 0.349 | 0.298 | 0.348 | 0.296 | 0.347 | **0.292** | **0.344** | 0.297 | 0.347 |
| | 192 | 0.375 | 0.398 | 0.376 | 0.396 | 0.378 | 0.396 | 0.387 | 0.401 | **0.373** | **0.394** |
| | 336 | 0.420 | 0.431 | 0.417 | 0.430 | 0.423 | 0.428 | 0.428 | 0.435 | **0.410** | **0.426** |
| | 720 | 0.433 | 0.448 | 0.423 | 0.442 | 0.421 | 0.435 | **0.409** | **0.433** | 0.411 | **0.433** |
| | Avg | 0.381 | 0.406 | 0.379 | 0.404 | 0.379 | 0.401 | 0.379 | 0.403 | **0.373** | **0.400** |
| PEMS04 | 12 | 0.084 | 0.189 | 0.075 | 0.177 | 0.078 | 0.182 | 0.077 | 0.180 | **0.074** | **0.176** |
| | 24 | 0.113 | 0.220 | 0.090 | 0.196 | 0.095 | 0.204 | 0.094 | 0.203 | **0.088** | **0.194** |
| | 48 | 0.164 | 0.266 | 0.117 | 0.225 | 0.126 | 0.236 | 0.120 | 0.231 | **0.110** | **0.219** |
| | 96 | 0.209 | 0.304 | 0.142 | 0.250 | 0.164 | 0.269 | 0.147 | 0.258 | **0.135** | **0.244** |
| | Avg | 0.143 | 0.245 | 0.106 | 0.212 | 0.116 | 0.223 | 0.109 | 0.218 | **0.102** | **0.208** |

## C.3 Full Results of STAR Ablation

The complete results of our ablation on universality of STAR are presented in Table 8. The STar Aggregate-Redistribute (STAR) module is a set-to-set function [41] that is replaceable to arbitrary transformer-based methods that use the attention mechanism. In this paragraph, we test the effectiveness of STAR on different existing transformer-based forecasters, such as PatchTST [29] and Crossformer [50]. Note that our method can be regarded as replacing the channel attention in iTransformer [27]. Here we involve substituting the time attention in PatchTST with STAR and incrementally replacing both the time and channel attention in Crossformer with STAR. The results, as presented in Table 8, demonstrate that replacing attention with STAR, which deserves less computational resources, could maintain and even improve the models' performance in several datasets.

## C.4 More Results of Lookback Ablation

In this section, we extend the lookback ablation in section 4.2 to $L \in [48, 720]$. Figure 7 shows the results in MSE. SOFTS performs almost consistently better than other models under different lookback window lengths. However, we also warn about the potential overfitting when the lookback length is very large, *i.e.* $L = 512$ or $L = 720$.

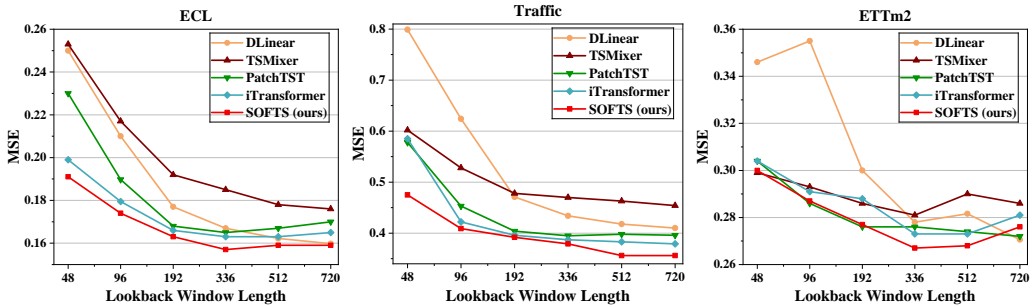

Figure 7: Influence of lookback window length $L \in \{48, 96, 192, 336, 512, 720\}$. SOFTS performs almost consistently better than other models under different lookback window lengths.

## C.5 Full Results of Hyperparameter Sensitivity Experiments

We investigate the impact of several key hyperparameters on our model's performance: the hidden dimension of the model, denoted as $d$, the hidden dimension of the core, represented by $d'$, and the number of encoder layers, $N$. Figure 8 and Figure 10 indicate that complex traffic datasets (such as Traffic and PEMS) require larger hidden dimensions and more encoding layers to handle their intricacies effectively. Moreover, Figure 9 shows that variations in $d'$ don't influence the model's overall performance so much.

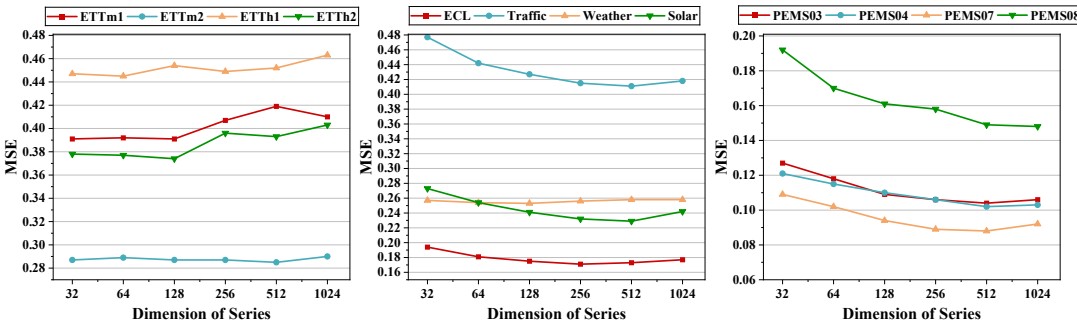

Figure 8: Influence of the hidden dimension of series $d$. Traffic datasets (such as Traffic and PEMS) require larger hidden dimensions to handle their intricacies effectively.

Table 8: The performance of STAR in different models. The attention replaced by STAR here are the time attention in PatchTST, the channel attention in iTransformer, and both the time attention and channel attention in modified Crossformer. The results demonstrate that replacing attention with STAR, which requires less computational resources, could maintain and even improve the models' performance in several datasets. [†]: The Crossformer used here is a modified version that replaces the decoder with a flattened head like what PatchTST does.

| Model | | iTransformer | | | | PatchTST | | | | Crossformer | | | |
|---|---|---|---|---|---|---|---|---|---|---|---|---|---|
| Component | | Attention | | STAR | | Attention | | STAR | | Attention | | STAR | |
| Metric | | MSE | MAE | MSE | MAE | MSE | MAE | MSE | MAE | MSE | MAE | MSE | MAE |
| Electricity | 96 | 0.148 | 0.240 | **0.143** | **0.233** | 0.164 | 0.251 | **0.160** | **0.248** | **0.156** | **0.259** | 0.166 | 0.263 |
| | 192 | 0.162 | 0.253 | **0.158** | **0.248** | 0.173 | 0.262 | **0.169** | **0.257** | 0.182 | 0.284 | **0.182** | **0.277** |
| | 336 | **0.178** | **0.269** | **0.178** | **0.269** | 0.190 | 0.279 | **0.187** | **0.275** | 0.203 | 0.305 | **0.200** | **0.296** |
| | 720 | 0.225 | 0.317 | **0.218** | **0.305** | 0.230 | 0.313 | **0.225** | **0.308** | 0.267 | 0.358 | **0.243** | **0.334** |
| | Avg | 0.178 | 0.270 | **0.174** | **0.264** | 0.189 | 0.276 | **0.185** | **0.272** | 0.202 | 0.301 | **0.198** | **0.292** |
| Traffic | 96 | 0.395 | 0.268 | **0.376** | **0.251** | 0.427 | 0.272 | **0.423** | **0.265** | **0.508** | **0.275** | 0.520 | 0.277 |
| | 192 | 0.417 | 0.276 | **0.398** | **0.261** | 0.454 | 0.289 | **0.434** | **0.271** | **0.519** | **0.281** | 0.535 | 0.285 |
| | 336 | 0.433 | 0.283 | **0.415** | **0.269** | 0.450 | 0.282 | **0.447** | **0.278** | 0.556 | 0.304 | **0.551** | **0.292** |
| | 720 | 0.467 | 0.302 | **0.447** | **0.287** | 0.484 | 0.301 | 0.489 | **0.301** | 0.600 | 0.329 | **0.591** | **0.315** |
| | Avg | 0.428 | 0.282 | **0.409** | **0.267** | 0.454 | 0.286 | **0.448** | **0.279** | 0.546 | 0.297 | 0.549 | **0.292** |
| Weather | 96 | 0.174 | 0.214 | **0.166** | **0.208** | 0.176 | 0.217 | **0.170** | **0.214** | **0.174** | 0.245 | **0.174** | **0.239** |
| | 192 | 0.221 | 0.254 | **0.217** | **0.253** | 0.221 | 0.256 | **0.215** | **0.251** | **0.219** | 0.283 | 0.220 | **0.282** |
| | 336 | **0.278** | **0.296** | 0.282 | 0.300 | 0.275 | **0.296** | **0.273** | **0.296** | **0.271** | 0.327 | 0.272 | **0.324** |
| | 720 | 0.358 | **0.347** | **0.356** | 0.351 | 0.352 | **0.346** | **0.349** | **0.346** | 0.351 | 0.383 | **0.343** | **0.376** |
| | Avg | 0.258 | 0.278 | **0.255** | **0.278** | 0.256 | 0.279 | **0.252** | **0.277** | 0.254 | 0.310 | **0.252** | **0.305** |
| PEMS03 | 12 | 0.071 | 0.174 | **0.064** | **0.165** | 0.073 | 0.178 | **0.071** | **0.173** | 0.067 | 0.170 | **0.065** | **0.165** |
| | 24 | 0.093 | 0.201 | **0.083** | **0.188** | 0.105 | 0.212 | **0.101** | **0.206** | 0.081 | 0.187 | **0.081** | **0.184** |
| | 48 | 0.125 | 0.236 | **0.114** | **0.223** | 0.159 | 0.264 | **0.157** | **0.256** | 0.109 | 0.220 | **0.109** | **0.216** |
| | 96 | 0.164 | 0.275 | **0.156** | **0.264** | 0.210 | 0.305 | **0.205** | **0.296** | 0.142 | 0.255 | 0.147 | **0.250** |
| | Avg | 0.113 | 0.222 | **0.104** | **0.210** | 0.137 | 0.240 | **0.134** | **0.233** | 0.100 | 0.208 | **0.100** | **0.204** |
| PEMS04 | 12 | 0.078 | 0.183 | **0.074** | **0.176** | 0.085 | 0.189 | **0.082** | **0.184** | **0.069** | **0.171** | 0.071 | 0.174 |
| | 24 | 0.095 | 0.205 | **0.088** | **0.194** | 0.115 | 0.222 | **0.108** | **0.214** | 0.082 | 0.190 | **0.079** | **0.185** |
| | 48 | 0.120 | 0.233 | **0.110** | **0.219** | 0.167 | 0.273 | **0.155** | **0.258** | 0.097 | 0.207 | **0.091** | **0.200** |
| | 96 | 0.150 | 0.262 | **0.135** | **0.244** | 0.211 | 0.310 | **0.198** | **0.297** | 0.111 | 0.222 | **0.106** | **0.218** |
| | Avg | 0.111 | 0.221 | **0.102** | **0.208** | 0.145 | 0.249 | **0.136** | **0.238** | 0.090 | 0.198 | **0.087** | **0.194** |
| PEMS07 | 12 | 0.067 | 0.165 | **0.057** | **0.152** | 0.068 | 0.163 | **0.065** | **0.160** | 0.056 | 0.151 | **0.055** | **0.150** |
| | 24 | 0.088 | 0.190 | **0.073** | **0.173** | 0.102 | 0.201 | **0.098** | **0.195** | 0.070 | 0.166 | **0.067** | **0.165** |
| | 48 | 0.110 | 0.215 | **0.096** | **0.195** | 0.170 | 0.261 | **0.162** | **0.250** | 0.090 | 0.192 | **0.088** | **0.183** |
| | 96 | 0.139 | 0.245 | **0.120** | **0.218** | 0.236 | 0.308 | **0.222** | **0.294** | 0.120 | 0.215 | **0.110** | **0.203** |
| | Avg | 0.101 | 0.204 | **0.087** | **0.184** | 0.144 | 0.233 | **0.137** | **0.225** | 0.084 | 0.181 | **0.080** | **0.175** |

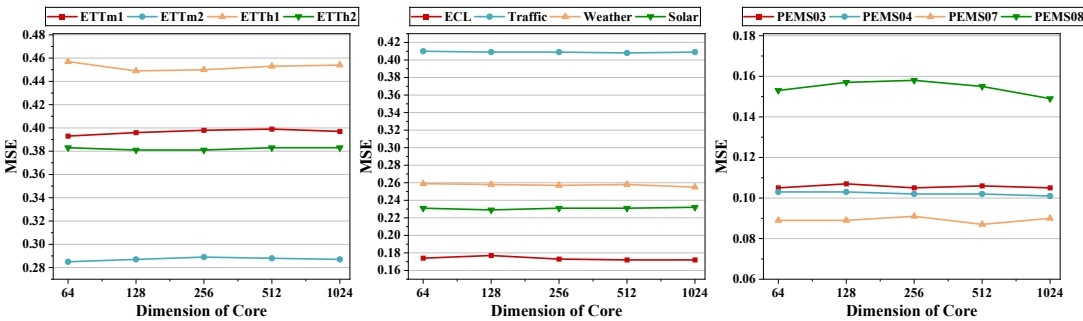

Figure 9: Influence of the hidden dimension of the core $d'$. Variations in $d'$ have a minimal influence on the model's overall performance

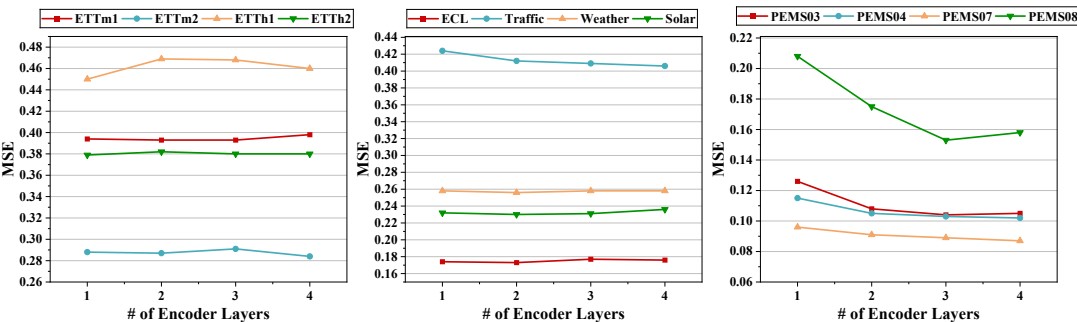

Figure 10: Influence of the number of the encoder layers $N$. Traffic datasets (such as Traffic and PEMS) require more encoding layers to handle their intricacies effectively.

## D   Error Bar

In this section, we test the robustness of SOFTS. We conducted 5 experiments using different random seeds, and the averaged results are presented in Table 9. It can be seen that SOFTS have robust performance over different datasets and different horizons.

Table 9: Robustness of SOFTS. Results are averaged over 5 experiments with different random seeds.

| Dataset | ETTm1 | | Weather | | Traffic | |
|---|---|---|---|---|---|---|
| Horizon | MSE | MAE | MSE | MAE | MSE | MAE |
| 96 | $0.325 \pm 0.002$ | $0.361 \pm 0.002$ | $0.166 \pm 0.002$ | $0.208 \pm 0.002$ | $0.376 \pm 0.002$ | $0.251 \pm 0.001$ |
| 192 | $0.375 \pm 0.002$ | $0.389 \pm 0.003$ | $0.217 \pm 0.003$ | $0.253 \pm 0.002$ | $0.398 \pm 0.002$ | $0.261 \pm 0.002$ |
| 336 | $0.405 \pm 0.004$ | $0.412 \pm 0.003$ | $0.282 \pm 0.001$ | $0.300 \pm 0.001$ | $0.415 \pm 0.002$ | $0.269 \pm 0.002$ |
| 720 | $0.466 \pm 0.004$ | $0.447 \pm 0.002$ | $0.356 \pm 0.002$ | $0.351 \pm 0.002$ | $0.447 \pm 0.002$ | $0.287 \pm 0.001$ |

| Dataset | PEMS03 | | PEMS04 | | PEMS07 | |
|---|---|---|---|---|---|---|
| Horizon | MSE | MAE | MSE | MAE | MSE | MAE |
| 12 | $0.064 \pm 0.002$ | $0.165 \pm 0.002$ | $0.074 \pm 0.000$ | $0.176 \pm 0.000$ | $0.057 \pm 0.000$ | $0.152 \pm 0.000$ |
| 24 | $0.083 \pm 0.002$ | $0.188 \pm 0.002$ | $0.088 \pm 0.000$ | $0.194 \pm 0.000$ | $0.073 \pm 0.003$ | $0.173 \pm 0.004$ |
| 48 | $0.114 \pm 0.004$ | $0.223 \pm 0.003$ | $0.110 \pm 0.001$ | $0.219 \pm 0.002$ | $0.096 \pm 0.002$ | $0.195 \pm 0.002$ |
| 96 | $0.156 \pm 0.001$ | $0.264 \pm 0.001$ | $0.135 \pm 0.003$ | $0.244 \pm 0.003$ | $0.120 \pm 0.003$ | $0.218 \pm 0.003$ |

# E    Showcase

## E.1    Visualization of the Core

In this section, we present a visualization of the core. The visualization is generated by employing a frozen state of our trained model to capture the series embeddings from the final encoder layer. These embeddings are then utilized as inputs to a two-layer MLP autoencoder. The primary function of this autoencoder is to map these high-dimensional embeddings back to the original input series. The visualization is shown in Figure 11. Highlighted by the red line, this core captures the global trend of all cross all the channels in

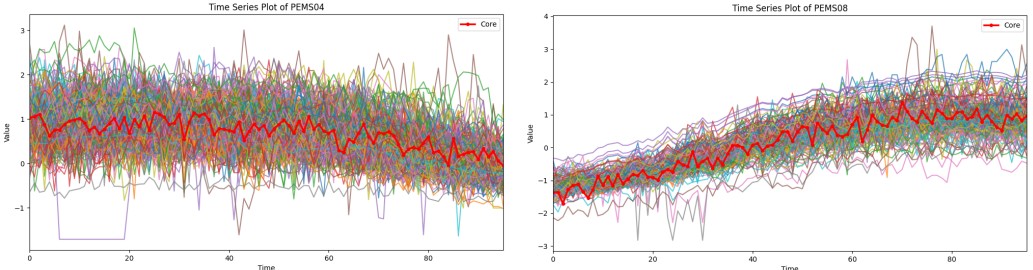

Figure 11: Visualization of the core, represented by the red line, alongside the original input channels. We freeze our model and extract the series embeddings from the last encoder layer to train a two-layer MLP autoencoder. This autoencoder maps the embeddings back to the original series, allowing us to visualize the core effectively.

## E.2    Visualization of Predictions

To provide a more intuitive demonstration of our model's performance, we present prediction showcases on the ECL (Figure 12), ETTh2 (Figure 13), Traffic (Figure 14), and PEMS03 (Figure 15) datasets. Additionally, we include prediction showcases from iTransformer and PatchTST on these datasets. The lookback window length and horizon are set to 96.

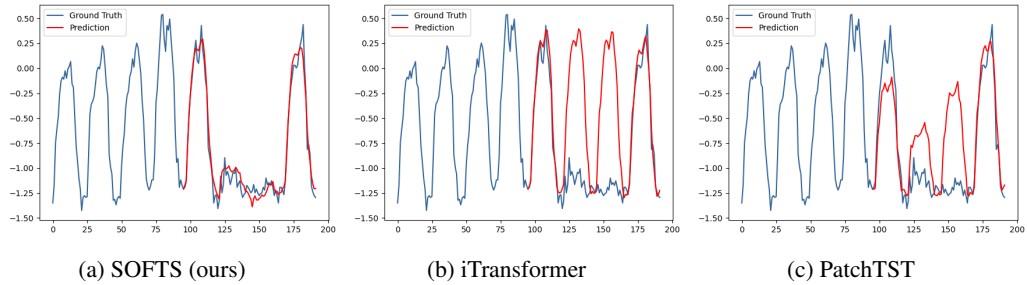

(a) SOFTS (ours)  (b) iTransformer  (c) PatchTST

Figure 12: Visualization of Prediction on ECL dataset with lookback window 96, horizon 96.

## E.3    More Results on Adaptation of Series Embedding

In this section, we show more results on the series embedding adaptation of our STAR module, similar to showcases in figure 6a and figure 6b. The number of channels should be large enough to show the relationship between channels in the embedding space. Therefore, we select the datasets ECL, PEMS03, and Traffic with channels 321, 358, and 862 respectively. Figure 16 shows the results on these datasets.

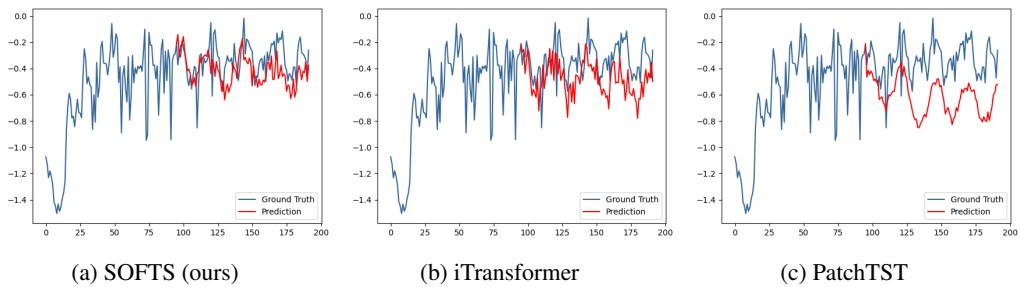

(a) SOFTS (ours)                (b) iTransformer                (c) PatchTST

Figure 13: Visualization of Prediction on ETTh2 dataset with lookback window 96, horizon 96.

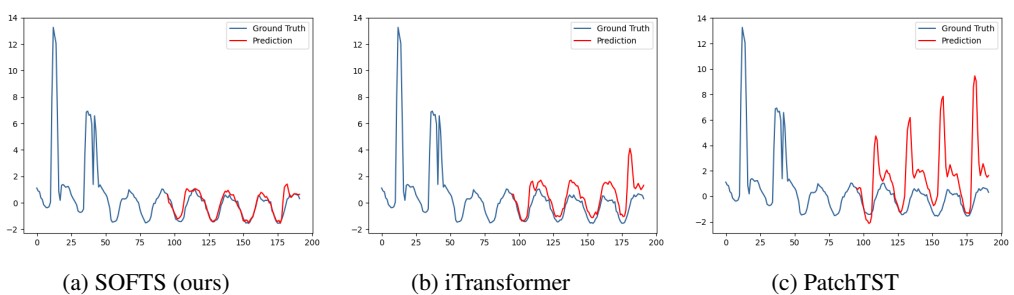

(a) SOFTS (ours)                (b) iTransformer                (c) PatchTST

Figure 14: Visualization of Prediction on Traffic dataset with lookback window 96, horizon 96.

### E.4 Visualization of Predictions on Abnormal Channels

As stated in Section 4.2, after being adjusted by STAR, the abnormal channels can be clustered towards normal channels by exchanging channel information. In this section, we choose two abnormal channels in the ECL and PEMS03 datasets to demonstrate our SOFTS model's advantage in handling noise from abnormal channels. As shown in Figure 17, the value of channel 160 in PEMS03 experiences a sharp decrease followed by a smooth period. In this case, SOFTS is able to capture the slowly increasing trend effectively. Similarly, in Figure 18, the signal of channel 298 in ECL resembles the sum of an impulse function and a step function, which lacks a continuous trend. Here, our SOFTS model provides a more stable prediction compared to the other two models.

## F   Limitations and Future Works

While the Series-cOre Fused Time Series (SOFTS) forecaster demonstrates significant improvements in multivariate time series forecasting, several limitations must be acknowledged, providing directions for future work.

**Dependence on core representation quality.**   The effectiveness of the STAR module heavily depends on the quality of the global core representation. If this core representation does not accurately capture the essential features of the individual series, the model's performance might degrade. Ensuring the robustness and accuracy of this core representation across diverse datasets remains a challenge that warrants further research.

**Limited exploration of alternative aggregate-redistribute strategies.**   Although the STAR module effectively aggregates and redistributes information, the exploration of alternative strategies is limited. Future work could investigate various methods for aggregation and redistribution to identify potentially more effective approaches, thereby enhancing the performance and robustness of the model.

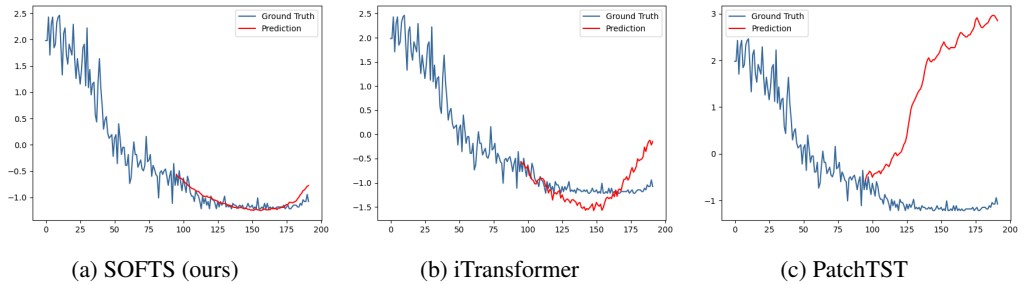

(a) SOFTS (ours)       (b) iTransformer       (c) PatchTST

Figure 15: Visualization of Prediction on PEMS03 dataset with lookback window 96, horizon 96.

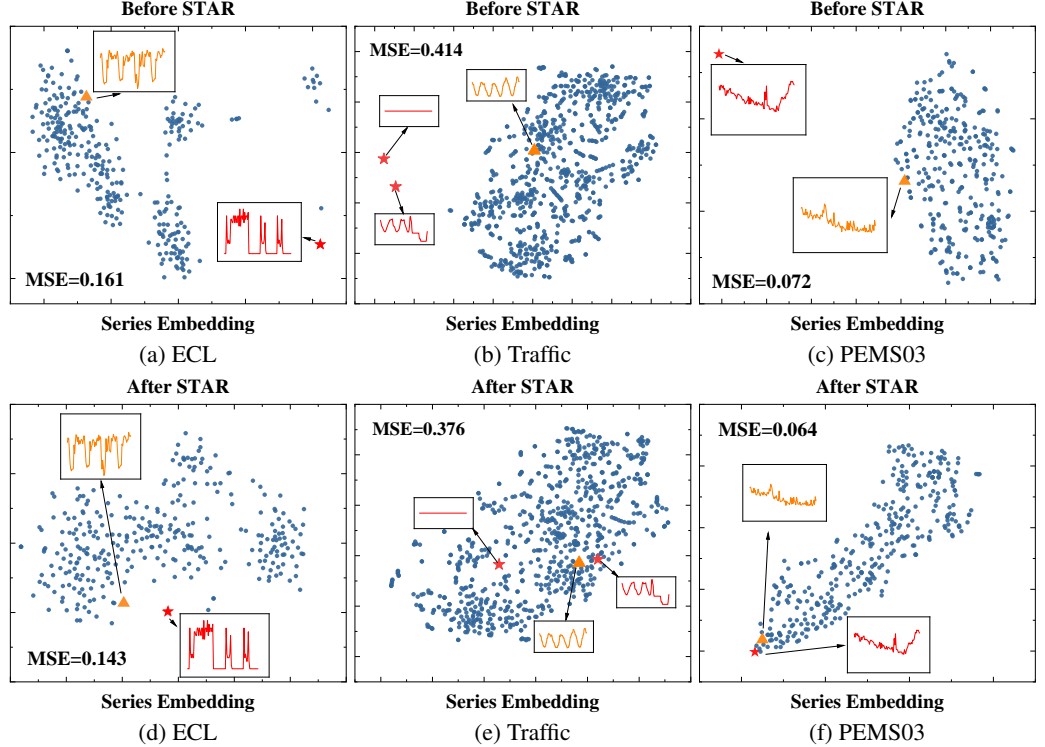

(a) ECL       (b) Traffic       (c) PEMS03

(d) ECL       (e) Traffic       (f) PEMS03

Figure 16: t-SNE visualization of series embeddings before and after STAR adjustment for ECL with a lookback window of 96 and horizon of 96, Traffic with a lookback window of 96 and horizon of 96 and for PEMS03 with a lookback window of 96 and horizon of 12. (a)-(d), (b)-(e), (c)-(f): The abnormal channel (⋆) is initially located far from the other channels. After adjustment by STAR, the abnormal channel clusters towards the normal channels (△) by exchanging channel information. Adapted series embeddings consistently improve forecasting performance based on the MSE metric.

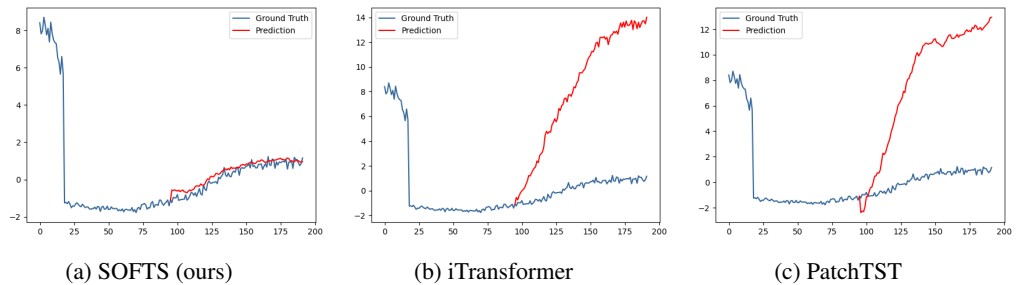

| (a) SOFTS (ours) | (b) iTransformer | (c) PatchTST |

Figure 17: Visualization of Prediction on abnormal channel in PEMS03 dataset with lookback window 96, horizon 96.

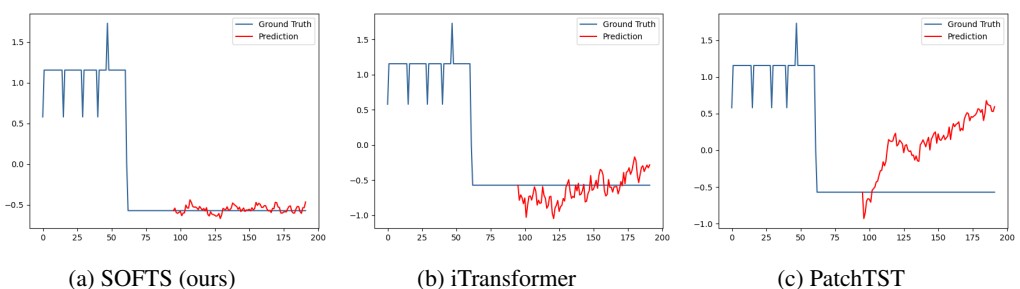

| (a) SOFTS (ours) | (b) iTransformer | (c) PatchTST |

Figure 18: Visualization of Prediction on abnormal channel in ECL dataset with lookback window 96, horizon 96.

## G   Societal Impacts

The development of the Series-cOre Fused Time Series (SOFTS) forecaster has the potential to significantly benefit various fields such as finance, traffic management, energy, and healthcare by improving the accuracy and efficiency of time series forecasting, thereby enhancing decision-making processes and optimizing operations. However, there are potential negative societal impacts to consider. Privacy concerns may arise from the use of personal data, especially in healthcare and finance, leading to possible violations if data is not securely handled. Additionally, biases in the data could result in unfair outcomes, perpetuating or exacerbating existing disparities. Over-reliance on automated forecasting models might lead to neglect of important contextual or qualitative factors, causing adverse outcomes when predictions are incorrect. To mitigate these risks, robust data protection protocols should be implemented, and continuous monitoring for bias is necessary to ensure fairness. Developing ethical use policies and maintaining human oversight in decision-making can further ensure that the deployment of SOFTS maximizes its positive societal impact while minimizing potential negative consequences.

