# OpenReview forum: "SOFTS: Efficient Multivariate Time Series Forecasting with Series-Core Fusion"
_NeurIPS.cc/2024/Conference — NeurIPS 2024 poster_

### Official Review · Reviewer_hzF8 · 2024-06-29

**Soundness:** 4
**Presentation:** 4
**Contribution:** 4
**Rating:** 7
**Confidence:** 4

**Summary:**

The paper introduces SOFTS (Series-cOre Fused Time Series forecaster), an efficient multivariate time series forecasting model that addresses the gap between channel independence and channel correlation in a novel way. By utilizing a centralized STAR (STar Aggregate-Redistribute) module, SOFTS creates a global core representation aggregated from all series, which is then redistributed and fused with individual series representations. This mechanism allows for efficient channel interaction while reducing dependence on the quality of each channel. The paper demonstrates SOFTS's superiority over state-of-the-art methods in terms of performance and computational complexity, and showcases the STAR module's adaptability across various forecasting models.

**Strengths:**

- SOFTS proposes a unique approach to handling channel correlations in multivariate time series forecasting by employing a centralized STAR module, which aggregates and redistributes series representations efficiently.
- The model's design and implementation are well-thought-out, and the empirical results show that SOFTS outperforms existing state-of-the-art methods.
- The paper is well-written and structured, providing clear explanations of the STAR mechanism and its integration into SOFTS.
- The proposed method has the potential to enhance forecasting accuracy and efficiency across various domains, making it a valuable addition to the field.

**Weaknesses:**

- Additional experiments evaluating the robustness of SOFTS under varying conditions of distribution drift could provide deeper insight into its reliability.

**Questions:**

- Has the author currently tried any other methods for aggregation and redistribution?
- SOFTS is a linear based model, so how to handle non-linear time series prediction?

**Limitations:**

- The authors have adequately addressed the technical limitations of their work within the paper.

---

> ### Author Rebuttal · Authors · 2024-08-06
>
> Thank you for your positive assessment of our work and your thoughtful feedback. We are pleased to know that you found our approach and presentation to be excellent and that you see the value of our contributions to the field. We have carefully considered your comments and here are our responses.
>
>
> > Additional experiments evaluating the robustness of SOFTS under varying conditions of distribution drift could provide deeper insight into its reliability.
>
> Thanks for your advice. We have tested the robustness of SOFTS on real-world datasets and synthetic data with manually injected noise in Figure 6. More rigorous experiments need further design and we are working on it. Thanks again for your valuable advice.
>
> > Has the author currently tried any other methods for aggregation and redistribution
>
> To keep our method neat, we use the simplest redistribution by concatenation and MLP fusion. We also experiment with 4 simple variants of aggregation, including mean pooling, max pooling, stochastic pooling, and weighted average as shown in Table 3. This simple structure is found to perform well. We agree that there would be other better designs for aggregation and redistribution. And we leave it for future exploration.
>
> > SOFTS is a linear based model, so how to handle non-linear time series prediction?
>
> The encoding part of SOFTS (using STAR) is composed of multiple MLP layers. So it can handle non-linear time series prediction.

---

> > ### Comment · Reviewer_hzF8 · 2024-08-13
> >
> > Thank you for your detailed rebuttal. The additional experiments on robustness and the clarification on aggregation methods are appreciated. The explanation of SOFTS's capability to handle non-linear predictions is satisfactory. We look forward to seeing the further refined experiments in the final version. Well done!

---

> > > ### Author Response · Authors · 2024-08-13
> > >
> > > Thanks very much for your recognition. We are happy that your concerns have been fully resolved.

---

### Official Review · Reviewer_NURe · 2024-07-07

**Soundness:** 3
**Presentation:** 3
**Contribution:** 3
**Rating:** 5
**Confidence:** 4

**Summary:**

This paper presents an efficient MLP-based model, the Series-cOre Fused Time Series forecaster (SOFTS). SOFTS incorporates a novel STar Aggregate-Redistribute (STAR) module to aggregate all series to form a global core representation, which is then dispatched and fused with individual series representations to facilitate channel interactions. The broad applicability of the STAR module across different forecasting models is also demonstrated empirically.

**Strengths:**

1. The paper is well-written.
2. The proposed STAR module is designed as a core to aggregate and exchange information from the channels efficiently, which is a universal module and can replace the attention mechanism.
3. The experiments in Figure 6 are interesting and the results are impressive.

**Weaknesses:**

The paper lacks significant innovation. The stochastic pooling itself is not novel in deep neural networks. A somewhat novel facet of the proposed model is to use the pooling to extract global representations. However, the idea of aggregate-and-dispatch the interactions between variables has already been studied in TimeXer [1].


[1] Wang, Y., Wu, H., Dong, J., Liu, Y., Qiu, Y., Zhang, H., ... & Long, M. (2024). Timexer: Empowering transformers for time series forecasting with exogenous variables. arXiv preprint arXiv:2402.19072.

**Questions:**

1. It seems that all series share a core representation and MLP layers, but different variables will require different aspects of information. How can the model address these differences?
2. Please increase the look-back window length of the input series to  512 and 720, and compare the performance with other baseline models.
3. Please provide more results on abnormal channels, i.e. ETT, ECL, and Traffic dataset.

**Limitations:**

yes

---

> ### Author Rebuttal · Authors · 2024-08-06
>
> Thank you for your positive feedback and constructive comments on our paper. We appreciate your recognition of our work's strengths and have carefully considered your suggestions for improvement. Below, we provide detailed responses to each of your points.
>
> > The paper lacks significant innovation. The stochastic pooling itself is not novel in deep neural networks. A somewhat novel facet of the proposed model is to use the pooling to extract global representations. However, the idea of aggregate-and-dispatch the interactions between variables has already been studied in TimeXer [1].
>
> Thanks very much for reminding us of this paper. We understand your concerns regarding the novelty of our method. However, our approach differs significantly from TimeXer. Most importantly, TimeXer still uses attention (self-attention, cross-attention) as the module for extracting correlations between tokens. In contrast, the core innovation of our paper, the STAR module (as emphasized in line 10 of the abstract, line 63 of the introduction, and line 126 of Section 3.2), is distinct from attention. This difference is detailed in Section 3.2 and illustrated in Figure 2. Additionally, TimeXer aggregates the information of series and patch embeddings using attention, which is an aggregation in the time dimension. This method does not address the problem our paper focuses on, which is how to efficiently and robustly model channel correlation. Finally, TimeXer investigates the scenario of forecasting with exogenous variables, which differs from our study of multivariate forecasting. Their proposed method is also tailored for that scenario. **Therefore, in terms of application scenarios, problems to solve, and methodology, there is little overlap between our paper and TimeXer. We believe that our work offers sufficient innovation compared to this paper.**
>
> In conclusion, we sincerely thank you for mentioning this paper and offering your suggestions. To ensure the completeness of our paper, we will add a discussion on the scenarios and methods of TimeXer in the related work and future work sections. The preliminary content is as follows:
>
> **Related Work:**
> TimeXer [1] uses self-attention to aggregate the information of series and patches and employs cross-attention to achieve interaction between endogenous and exogenous variables. It extends iTransformer to the forecasting with exogenous variables scenario. Unlike these two methods, our paper proposes an efficient channel interaction module based on MLP, achieving better performance with lower complexity.
>
> **Future Work:**
> Future work includes exploring how to apply SOFTS and STAR to more forecasting scenarios, such as forecasting with exogenous variables [1] and forecasting with future variates.
>
> [1] Wang, Y., Wu, H., Dong, J., Liu, Y., Qiu, Y., Zhang, H., ... & Long, M. (2024). Timexer: Empowering transformers for time series forecasting with exogenous variables. arXiv preprint arXiv:2402.19072.
>
> > It seems that all series share a core representation and MLP layers, but different variables will require different aspects of information. How can the model address these differences?
>
> The MLP module can fuse different parts of the core representation according to the series due to its universal approximation ability. That is $MLP(s,o)$ can approximate any forecasting function if $s,o$ contains enough information. By Kolmogorov-Arnold representation theorem and DeepSets (line 146 and Section B.2). $o$ with equation (3) can approximate the function on all the series. Therefore, even if different variables will require different aspects of information, STAR can also approximate the required function.
>
>
> > Please increase the look-back window length of the input series to 512 and 720, and compare the performance with other baseline models.
> > Please provide more results on abnormal channels, i.e. ETT, ECL, and Traffic dataset.
>
> Thanks for your advice. We have added the lookback window and the abnormal channel results (on ECL, Traffic, PEMS) in the **global rebuttal PDF**. The ETT dataset has only 7 channels, and the embedding correlation of the channels is not very clear, so we replaced it with PEMS. And we will make corresponding modifications to the revised version.

---

> > ### Comment · Reviewer_NURe · 2024-08-13
> >
> > Thank you for your rebuttal and for addressing my concerns. I still maintain my original score as a borderline accept. I appreciate using stochastic pooling to capture global core representation, while it lacks theoretical analysis. As described in $\underline{\text{Line 437-439}}$, the authors tested several common pooling methods, and stochastic pooling is more of a result-oriented design.

---

> > > ### Author Response · Authors · 2024-08-13
> > >
> > > Thank you very much for your reply. Stochastic pooling is indeed the best-performing design, but it is not the core contribution of this paper; it is merely an implementation method for the STAR structure we proposed. In most cases, the performance impact is primarily due to the STAR structure rather than the pooling method. Anyway, we appreciate your positive attitude toward our work.

---

### Official Review · Reviewer_mAnn · 2024-07-12

**Soundness:** 3
**Presentation:** 3
**Contribution:** 2
**Rating:** 5
**Confidence:** 5

**Summary:**

This paper proposes to use a global representation to capture the channel correlations for multivariate time series forecasting. Specifically, it uses stochastic pooling to get the global representation by aggregating representations of individual series and then concats the global representation and individual representations to reflect channel correlations for each series. Experiment results confirm that the proposal is much efficiency and achieves better performance than existing methods.

**Strengths:**

1. This paper proposes an efficient method to capture the channel correlations for multivariate time series forecasting.
2. Extensive experiments are conducted to confirm the effectiveness of the proposal.
3. The paper is well-written and easy to understand in general.

**Weaknesses:**

1. Some experimental results are unconvincing.

-It is unclear why different datasets and metrics are used for different ablation studies, e.g., the datasets used in Table 3 and 4 are different, MAE is used in Figure 4 but MSE is used in other figures.

-There is no statistical significance tests between the results of the proposal and baselines.

-The results of Lookback Window Length 720 should be given as done in other papers.

-It is better to give the training time for each methods as well.

2. Some places are not clearly described.

-It is unclear the MLP and Linear operations are channel independent or dependent. I can guess they are channel independent, but it is not clearly described in Figure 1 and Section 3.

-The sentence "... rely too heavily on the correlation to achieve satisfactory results under distribution drift" in the Abstract is not clearly explained.

3. Minor mistakes or typos: Embedding module is missing in Figure1; oi ∈ Rd' should be oi ∈ RC×d' in Algorithm 1.

**Questions:**

Please refer to the weaknesses.

**Limitations:**

Yes

---

> ### Author Rebuttal · Authors · 2024-08-06
>
> Thank you for your thorough review of our paper. We appreciate your feedback and have made corresponding responses to address your concerns.
>
> > It is unclear why different datasets and metrics are used for different ablation studies, e.g., the datasets used in Table 3 and 4 are different, MAE is used in Figure 4 but MSE is used in other figures.
>
> Datasets of Table 3,4 are selected due to space restriction. We provide full results in the **global rebuttal** part. We found that MSE lines of SOTA methods usually entangle with each other, therefore we display MAE. To alleviate concerns, we have also included performance curves using MSE in Figure 1 of the **global rebuttal PDF**.
>
>
> > Statistical significance tests.
>
> Thank you for pointing out the necessity of a significance test. Across 48 settings of the benchmark, our method outperforms 10 baseline methods in most cases (Table 6). Considering the diversity of the datasets (covering electricity, traffic, energy, and climate) and the diversity of the comparison methods (including linear, MLP, CNN, and Transformers), this sufficiently demonstrates the superiority of our proposed method. We also demonstrate the stability under different random seeds in Table 9, indicating that the performance is not coincidental. To alleviate any concerns, we have supplemented our analysis with t-test significance experiments. Due to the large number of experimental settings and the lengthy training time of some methods, we selected iTransformer and PatchTST, which have performances most comparable to ours, for comparison. We repeat 10 times for each method and each setting. The results are as follows. The T-Statistics < 0 and p-value < 0.05 are marked in bold.
>
> |||PatchTST||||iTransformer||||
> |---|:---:|:---:|:---:|:---:|:---:|:---:|:---:|:---:|:---:|
> |||MAE||MSE||MAE||MSE||
> |||P-Value|T-Statistics|P-Value|T-Statistics|P-Value|T-Statistics|P-Value|T-Statistics|
> |ETTm2|96|0.853|0.191|0.514|**-0.683**|**0.001**|**-4.874**|**0.001**|**-5.199**|
> ||192|0.882|**-0.154**|0.748|**-0.332**|**0**|**-7.533**|**0**|**-9.142**|
> ||336|0.063|**-2.156**|0.319|**-1.063**|**0**|**-7.514**|**0.003**|**-4.169**|
> ||720|0.38|**-0.93**|0.925|**0.098**|**0.021**|**-2.877**|**0.108**|**-1.809**|
> |PEMS08|12|**0**|**-27.817**|**0**|**-42.337**|**0**|**-15.018**|**0**|**-21.369**|
> ||24|**0**|**-32.518**|**0**|**-43.611**|**0**|**-17.864**|**0**|**-20.731**|
> ||48|**0**|**-33.739**|**0**|**-38.155**|**0**|**-19.017**|**0**|**-36.447**|
> ||96|**0**|**-72.713**|**0**|**-122.96**|**0**|**-15.827**|**0**|**-16.639**|
> |ECL|96|**0**|**-63.882**|**0**|**-66.26**|**0**|**-10.45**|**0**|**-7.251**|
> ||192|**0**|**-18.627**|**0**|**-25.142**|**0**|**-6.495**|**0.002**|**-4.479**|
> ||336|**0**|**-12.705**|**0**|**-18.651**|0.288|**-1.139**|0.551|0.623|
> ||720|**0**|**-14.689**|**0**|**-25.911**|0.733|**-0.353**|0.194|**1.419**|
> |Solar|96|**0.013**|**-3.2**|0.225|**-1.315**|**0.033**|**-2.57**|**0.611**|**-0.529**|
> ||192|**0**|**-6.401**|0.573|0.587|**0**|**-9.082**|**0.063**|**-2.159**|
> ||336|**0.001**|**-4.953**|0.23|**-1.3**|**0**|**-6.172**|0.141|**-1.635**|
> ||720|0.735|0.35|0.807|0.252|0.261|**-1.21**|0.68|**-0.428**|
> |Traffic|96|**0**|**-175.79**|**0**|**-117.36**|**0**|**-23.495**|**0**|**-15.247**|
> ||192|**0**|**-161.43**|**0**|**-146.71**|**0**|**-96.104**|**0**|**-32.583**|
> ||336|**0**|**-92.217**|**0**|**-112.43**|**0**|**-53.061**|**0**|**-15.972**|
> ||720|**0**|**-109.05**|**0**|**-81.909**|**0**|**-53.852**|**0**|**-19.795**|
>
> Our method significantly outperforms the previous SOTA methods in most cases.
>
> > The results of Lookback Window Length 720.
>
> We extend the lookback window analysis from range [48, 336] to  [48, 720] and show the results in Figure 1 of the **global rebuttal PDF**. The figure shows that SOFTS can also achieve superior performance when the lookback window length is extended to 512 and 720.
>
>
> > Training time for each method.
>
> Thank you very much for your suggestion. However, training time is influenced not only by the complexity of the model itself but also by the training strategy. Earlier methods, such as FEDformer and Informer, often completed training early due to early stopping strategies to prevent overfitting. These methods have shorter training times than current SOTA methods but perform much worse. To ensure a fair comparison, we comprehensively present model performance, inference time, and memory in Figure 3, where the latter two are only related to the complexity of the model. Therefore, we believe this figure more fairly demonstrates the superiority of our method in terms of performance and efficiency compared to merely showing training time.
>
> > It is unclear the MLP and Linear operations are channel independent or dependent.
>
> Yes. To make it clear in the paper, we have specified every MLP in the form of mapping. For example, $\operatorname{MLP}_1: R^{d} \mapsto R^{d'}$ in line 147 means it project the axis with dimension $d$ to dimension $d'$. This mapping is shared across other axes including the channel dimension. Therefore, it corresponds to channel independence. They function similarly to the FFN layer of the Transformer.
>
> > The sentence "... rely too heavily on the correlation to achieve satisfactory results under distribution drift" in the Abstract is not clearly explained.
>
> This sentence corresponds to lines 33-35 in the introduction "However, such channel mixing structures were found vulnerable to the distribution drift". For rigor, we change it to "fail to achieve satisfactory results under distribution drift when incorporating the correlation".
>
> > Minor mistakes or typos: Embedding module is missing in Figure1; oi ∈ Rd' should be oi ∈ RC×d' in Algorithm 1.
>
> Thanks. We omitted the embedding module in the figure to spare space, and we will add it to the figure in the final version; $o_{i}$ is the global core representation for the whole multivariate series. It is not channel-specific, so it should be $o_{i} ∈ R^{d'}$ and it is not a typo.

---

> > ### Comment · Reviewer_mAnn · 2024-08-09
> >
> > Thank you for the responses. Considering other reviewers' comments as well, I have updated my rating.

---

> > > ### Author Response · Authors · 2024-08-09
> > >
> > > Thank you for raising the score. If you have any additional concerns or questions, we are willing to address them.

---

### Official Review · Reviewer_gtby · 2024-07-12

**Soundness:** 3
**Presentation:** 3
**Contribution:** 3
**Rating:** 7
**Confidence:** 3

**Summary:**

The authors present a framework for modeling correlations between channels in a multivariate time series forecasting task. This framework concatenates each channel embedding with a ‘global core embedding’ which contains information from all channels in the lookback window. The authors present experiments that demonstrate the utility of this concept, both from a performance and efficiency perspective.

**Strengths:**

Time series forecasting has been an important problem and it continues to grow with the advent of time series foundation models. To the best of my knowledge, it remains an open question for how to best enable multivariate time series forecasting, and this paper provides a conceptually reasonable approach. I believe the authors’ work is of broad interest.

**Weaknesses:**

In my opinion, the paper would be improved with analysis and discussion of their results.  There is little discussion beyond drawing attention to features of figures and tables, which misses an opportunity to explain why the authors believe they are observing such behavior.  I am specifically interested in a discussion between iTransformer and SOFTS, which appear to be quite similar along many dimensions that SOFTS claims to be superior.  Such a discussion will help guide potential readers through the considerations they should take into account when deciding which framework to implement on their own forecasting problems.

I also recommend adding either a few sentences or a small figure that highlights the differences between PatchTST, transformer, and SOFTS — I see some details in the text of table 4, but I think making this information more prominent would be helpful and make the paper more self-contained.

**Questions:**

1. Under what conditions should I choose SOFTS over iTransformer?  Figure 3a makes it appear as though when there are a sufficiently large number of channels in the time series.  Are there other considerations?  Figures 3b and 6c show slight improvements but the difference doesn't seem large enough to discriminate between which framework would be better for a reader's specific time series forecasting efforts.
2. What are the limitations of SOFTS? Or what are the tradeoffs between using SOFTS vs other models?  A few words of discussion in the conclusion would help improve the clarity of the paper.

**Limitations:**

As is, there is no discussion of the limitations in the main text of the paper.  To address this partially, section G of the appendix could be moved to the main text.

However, the discussion of the limitations of SOFTS currently exists in a vacuum, lacking a comparison/contrast with other models mentioned in the paper, especially PatchTST and iTransformer.  The authors should characterize and highlight the characteristics of datasets and inference tasks where SOFTS outperforms existing methods by wide margin.

---

> ### Author Rebuttal · Authors · 2024-08-06
>
> Thank you for taking the time to review our paper and for your constructive feedback. We appreciate your recognition of the importance of time series forecasting and your acknowledgment of our work's potential impact. We have carefully considered your comments and suggestions and have made revisions to the manuscript to address your concerns. Here are our responses.
>
> > Comparison and selection among PatchTST, Transformer, and SOFTS.
>
> We appreciate your suggestion to provide a more in-depth analysis and discussion of our results. The main difference between the three methods is in the following table.
>
> - Embedding: The way of creating token embeddings. "Patch" means tokens are embedded by small patches (or windows) containing values of consecutive time. "Series" means embedding the whole series.
>
> - Temporal: The way of extracting the correlation of different time steps.
>
>  - Channel: The way of extracting the correlation of different channels.
>
> | Method    | PatchTST  | iTransformer | SOFTS  |
> | --------- | --------- | ------------ | ------ |
> | Embedding | Patch     | Series       | Series |
> | Temporal  | Attention | MLP          | MLP    |
> | Channel   | /         | Attention    | STAR   |
>
> **PatchTST vs SOFTS**
>
> The main difference between PatchTST and SOFTS lies in their approach to handling channels. PatchTST uses a channel-independent model, which sacrifices channel correlation but eliminates interference from other channels in the prediction process. In contrast, our SOFTS model employs the STAR module to extract channel correlations more robustly and effectively while reducing interference from other channels. By adjusting the representation through channel clustering, as shown in Figure 6, the predictions become more robust, making SOFTS particularly advantageous when there are many channels. Although SOFTS minimizes the negative impact of channel correlation as much as possible, there is still a risk of overfitting in situations where channels are highly independent. Current multivariate datasets may not be so independent so we find SOFTS outperform in most cases.
>
> The second difference is the patch embedding of PatchTST, which utilizes a sliding window to extract information within consecutive times. PatchTST then exchanges the information at different time windows by attention. Our SOFTS uses MLP which is simpler but achieves SOTA performance as well.
>
> **iTransformer vs SOFTS**
>
> The main difference between iTransformer and SOFTS is that iTransformer uses attention for channel interaction, while SOFTS uses STAR. As shown in Figure 2, the significant difference is that STAR employs a centralized structure to enhance robustness, avoiding the influence of certain abnormal channels on the predictions of other channels. From Table 2, it is evident that SOFTS has a clear advantage over iTransformer on datasets with a large number of channels, such as Traffic and PEMS. This is likely because, with more channels, attention is more susceptible to the influence of certain abnormal channels, whereas STAR mitigates this effect through aggregation. Additionally, in terms of efficiency, STAR reduces complexity from quadratic to linear, significantly enhancing its scalability. But we also note that SOFTS may have a bottleneck due to interaction through core representation.
>
> **So, under what conditions should I choose SOFTS over iTransformer?**
> Based on the current results, we find that SOFTS generally outperforms iTransformer in time series problems, both in terms of performance and efficiency, especially when the number of channels is large. This might be because robustness considerations outweigh capacity in time series issues. Of course, we do not rule out the possibility of scenarios where the reverse might be true in the future.
>
> > Limitation of SOFTS
>
> We acknowledge your suggestion to move the discussion of limitations from the appendix to the main text. In response, we have integrated Section G from the appendix into the main body of the paper. We have also enhanced this section by comparing and contrasting the limitations of SOFTS with those of other models, such as PatchTST and iTransformer, like the discussions stated above.
>
> Once again, we sincerely thank you for your valuable feedback. Your insights have significantly contributed to improving the clarity and impact of our paper. We hope the revised manuscript addresses your concerns and enhances the overall quality of our work.

---

> > ### Comment · Reviewer_gtby · 2024-08-13
> >
> > Thank your for your response and your elaboration on some of the questions I had.  In my opinion, your revisions make the paper more broadly accessible and self contained. This should help amplify the impact of your work.  I have raised my score.  Nice work!

---

> > > ### Author Response · Authors · 2024-08-13
> > >
> > > Thanks very much for your recognition and valuable advice for improving our manuscript. We hope our work is helpful to you.

---

### Author Rebuttal · Authors · 2024-08-07

Dear Reviewers,

We sincerely appreciate the time and effort you have dedicated to reviewing our paper and for providing valuable feedback. We are delighted that the majority of the reviewers (3 out of 4) have given positive evaluations of our work. Our work is said to "of broad interest" (gtby), "well-thought-out" and "valuable addition to the field" (hzF8), "interesting" and "impressive" (NURe), "well-written" (mAnn,NURe,hzF8) . One reviewer has raised some concerns regarding the experiments and clarifications of our paper. We hope that this rebuttal addresses these concerns effectively, allowing both reviewers and readers to gain a clearer and more comprehensive understanding of our research. We aim to improve the perception and evaluation of our work through these clarifications and enhancements.

**PDF content**: Two reviewers (**mAnn, NURe**) are interested in the performance (in MSE) when the look-back window is 512 and 720. Figure 1 in the pdf displays the comprehensive performance comparison against other methods with look-back lengths in [48, 720]. Figure 2 shows more results on abnormal channels, in which reviewer **NURe** is interested.

Reviewer **mAnn** is also concerned about different datasets used in Table 3, 4. The datasets are selected due to the space restriction. Due to the character restriction of personal rebuttal, we show the full results here:

**Table 3. Comparison of the effect of different pooling methods. The term "w/o STAR" refers to a scenario where an MLP is utilized with the Channel Independent (CI) strategy, without the use of STAR. The result reveals that incorporating STAR into the model leads to a consistent enhancement in performance across all pooling methods. Apart from that, stochastic pooling performs better than mean and max pooling.**
|Pooling Method|**ECL**||**Traffic**||**Weather**||**Solar**||**ETTh2**||**PEMS03**||**PEMS04**||**PEMS07**||
|:-------------|:---------|:---------|:----------|:---------|:----------|:---------|:---------|:---------|:---------|:---------|:---------|:---------|:---------|:---------|:---------|:---------|
||**MSE**|**MAE**|**MSE**|**MAE**|**MSE**|**MAE**|**MSE**|**MAE**|**MSE**|**MAE**|**MSE**|**MAE**|**MSE**|**MAE**|**MSE**|**MAE**|
|w/oSTAR|0.187|0.273|0.442|0.281|0.261|0.281|0.247|0.272|0.381|0.406|0.135|0.235|0.143|0.245|0.143|0.232|
|Mean|**0.174**|0.266|0.420|0.277|0.261|0.281|0.234|0.262|0.379|0.404|0.106|0.212|0.106|0.212|0.090|0.188|
|Max|0.180|0.270|**0.406**|0.271|0.259|0.280|0.246|0.269|0.379|0.401|0.113|0.221|0.116|0.223|0.096|0.198|
|Weighted|0.184|0.275|0.440|0.292|0.263|0.284|0.264|0.280|0.379|0.403|0.118|0.226|0.109|0.218|0.097|0.200|
|Stochastic|**0.174**|**0.264**|0.409|**0.267**|**0.255**|**0.278**|**0.229**|**0.256**|**0.373**|**0.400**|**0.104**|**0.210**|**0.102**|**0.208**|**0.087**|**0.184**|

**Table 4. The performance of STAR in different models. The attention replaced by STAR here are the time attention in PatchTST, the channel attention in iTransformer, and both the time attention and channel attention in modified Crossformer.
The results demonstrate that replacing attention with STAR, which requires less computational resources, could maintain and even improve the models' performance in most datasets.**
|Model|Component|**ECL**||**Traffic**||**Weather**||**Solar**||**ETTh2**||**PEMS03**||**PEMS04**||**PEMS07**||
|---------------------|---------|---------|---------|-----------|-----------|-----------|-----------|---------|---------|---------|---------|----------|----------|----------|----------|----------|----------|
|||**MSE**|**MAE**|**MSE**|**MAE**|**MSE**|**MAE**|**MSE**|**MAE**|**MSE**|**MAE**|**MSE**|**MAE**|**MSE**|**MAE**|**MSE**|**MAE**|
|PatchTST|Attention|0.189|0.276|0.454|0.286|0.256|0.279|0.236|0.266|**0.385**|**0.410**|0.137|0.240|0.145|0.249|0.144|0.233|
||STAR|**0.185**|**0.272**|**0.448**|**0.279**|**0.252**|**0.277**|**0.231**|**0.259**|0.391|0.413|**0.134**|**0.233**|**0.136**|**0.238**|**0.137**|**0.225**|
|Crossformer|Attention|0.202|0.301|**0.546**|0.297|0.254|0.310|0.206|0.258|2.772|1.271|0.100|0.208|0.090|0.198|0.084|0.181|
||STAR|**0.198**|**0.292**|0.549|**0.292**|**0.252**|**0.305**|**0.200**|**0.252**|**1.919**|**1.043**|**0.100**|**0.204**|**0.087**|**0.194**|**0.080**|**0.175**|
|iTransformer|Attention|0.178|0.270|0.428|0.282|0.258|0.278|0.233|0.262|0.383|0.407|0.113|0.221|0.111|0.221|0.101|0.204|
||STAR|**0.174**|**0.264**|**0.409**|**0.267**|**0.255**|**0.278**|**0.229**|**0.256**|**0.373**|**0.400**|**0.104**|**0.210**|**0.102**|**0.208**|**0.087**|**0.184**|

Thank you once again for your constructive comments and support.

---

### Decision · Program_Chairs · 2024-09-25

**Decision:**

Accept (poster)

**Comment:**

This paper has been assessed by four knowledgeable reviewers. Two of them recommended accepting it (straight accept ratings) while two opted weakly for rejection. It presents an approach to modeling correlation between channels of multivariate time series to support forecasting tasks. The authors responded to initial feedback by providing a comprehensive rebuttal and engaged the reviewers in a  discussion. That resulted in improved understanding of the key points made by the authors and in increasing the reviewers' scores. This work is mature enough and sufficiently well presented to be included in the program of NeurIPS.